# NIMO: A NONLINEAR INTERPRETABLE MODEL

**Shijian Xu**
University of Basel
`shijian.xu@unibas.ch`

**Marcello Massimo Negri**
University of Basel
`marcellomassimo.negri@unibas.ch`

**Volker Roth**
University of Basel
`volker.roth@unibas.ch`

## ABSTRACT

Deep learning has achieved remarkable success across many domains, but it has also created a growing demand for interpretability in model predictions. Although many explainable machine learning methods have been proposed, post-hoc explanations lack guaranteed fidelity and are sensitive to hyperparameter choices, highlighting the appeal of inherently interpretable models. For example, linear regression provides clear feature effects through its coefficients. However, such models are often outperformed by more complex neural networks (NNs) that usually lack inherent interpretability. To address this dilemma, we introduce NIMO, a framework that combines inherent interpretability with the expressive power of neural networks. Building on the simple linear regression, NIMO is able to provide flexible and intelligible feature effects. Relevantly, we develop an optimization method based on parameter elimination, that allows for optimizing the NN parameters and linear coefficients effectively and efficiently. By relying on adaptive ridge regression we can easily incorporate sparsity as well. We show empirically that our model can provide faithful and intelligible feature effects while maintaining good predictive performance.

## 1 INTRODUCTION

Over the past decade, neural networks have achieved remarkable success across domains, from computer vision (Krizhevsky et al., 2012) to natural language processing (Vaswani et al., 2017). At the same time, their deployment in high-stakes domains such as healthcare has created a pressing demand for interpretability (Esteva et al., 2017). However, neural networks in general lack interpretability of the predictions in terms of the input features, hence the term "black-box" models. In contrast, classical models such as linear regression and decision trees are often considered as highly interpretable because of their transparent structure and simple decision rules, but they can lack predictive power on complex, high-dimensional, or highly nonlinear tasks (Hastie et al., 2009; Breiman et al., 2017). This tension between accuracy and interpretability has motivated much recent work on models and methods that combine the expressive power of neural networks with the transparency of simpler models (Lemhadri et al., 2021; Thompson et al., 2023; Wu et al., 2017; 2021). Meanwhile, many model-agnostic, post-hoc explainers have been proposed, such as SHAP (Lundberg & Lee, 2017) and LIME (Ribeiro et al., 2016), providing instance-level feature attributions for arbitrary predictors. But these post-hoc explanations are approximations and may depend on the choices of parameters, such as the background distribution and kernel width. Among various ways to achieve interpretability, feature effects (Scholbeck et al., 2019) provide a direct, quantitative description of how each feature influences the prediction, something post-hoc methods can approximate but do not guarantee.

We use the term feature effects in a broad sense, referring to how changes in input features influence model predictions. A common formalization of this idea is through marginal effects (Nguyen, 2020), which define feature effects as the direction and magnitude of the change in prediction due to an infinitesimal change in a feature value. Linear models offer immediately intelligible marginal effects through their coefficients (Molnar, 2025), but are outperformed by more complex nonlinear models, particularly neural networks. Unfortunately, these more expressive models typically lose the property

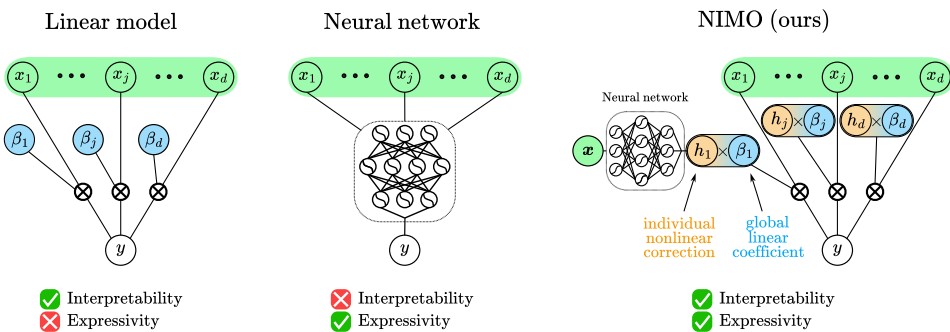

Figure 1: Comparison of linear models, neural networks, and our method. Linear models offer straightforward interpretability but limited expressivity while neural networks provide high expressivity but lack interpretability. Our method combines the strengths of both by building on the linear model and adjusting the global linear coefficients with individual nonlinear corrections.

of providing such inherently intelligible effects. In such models, marginal effects must instead be obtained by computing derivatives of the fitted function with respect to the inputs, usually through computationally expensive automatic differentiation (autograd) frameworks (Paszke et al., 2017). To bridge this gap, we propose an intrinsically interpretable hybrid model that preserves intelligible marginal effects while retaining the expressivity of neural networks. Specifically, we start from a linear model $y = \sum_j x_j \beta_j$, and adjust each $\beta_j$ by a multiplicative factor $h_j$, which is the output of a neural network $\text{NN}_{\boldsymbol{u}_j}(\mathbf{x})$ evaluated at the input instance $\mathbf{x}$:

$$y = \sum_{j=1}^{p} x_j \beta_j h_j(\mathbf{x}) \quad \text{with} \quad h_j(\mathbf{x}) = \text{NN}_{\boldsymbol{u}_j}(\mathbf{x}) \ . \tag{1}$$

In Figure 1, we illustrate how the proposed approach retains the high-level structure of a linear model while integrating per-instance nonlinear corrections through neural networks.

Within this framework, we distinguish between local explanations and global interpretations. Local explanations describe how changes in a feature influence the prediction for a specific instance, enabled here by the per-instance corrections $h_j(\mathbf{x})$ from the neural network. Global interpretations, in contrast, summarize the overall influence of a feature across the dataset, which we later formalize through the marginal effects at the mean (MEM). In our model, these coincide exactly with the global linear coefficients. To illustrate, consider a medical application where we aim to estimate a patient's risk of developing a disease. A local explanation would ask: *given this patient's age, weight, and other characteristics, how does a small change in age affect their individual risk?* A global interpretation instead asks: *if all other variables are held fixed, how does age in general influence disease risk across the population?* For linear models, these two perspectives coincide, but in our framework they differ due to potential nonlinear feature interactions. Our network is designed so that the model preserves the classical interpretation of the linear coefficients $\boldsymbol{\beta}$, while simultaneously providing per-instance corrections through $h_j(\mathbf{x})$. This enables us to unify global summaries via MEM with local explanations at the instance level within a single coherent framework.

Since the network parameters $\boldsymbol{u}$ and the linear coefficients $\boldsymbol{\beta}$ are tightly coupled, jointly optimizing them is non-trivial. To address this, we develop an optimization method based on parameter elimination. Specifically, we derive a closed-form expression for the linear coefficients $\boldsymbol{\beta}$ as a function of the NN parameters $\boldsymbol{u}$. Substituting this expression back into the objective reduces the problem to optimizing only over the NN parameters. To further enhance interpretability, we impose sparsity by formulating the problem as adaptive ridge regression (Grandvalet, 1998). In the network, we further apply group $\ell_2$ regularization (Yuan & Lin, 2006) to the weight matrix of the first fully connected layer, encouraging feature-level sparsity and clarifying how inputs contribute to predictions.

Overall, the contributions of the present work can be summarized as follows:

- We introduce a nonlinear interpretable model that combines the expressivity of neural networks with the interpretability of linear models, providing intelligible global interpretability via marginal effects at the mean (MEM) and per-instance corrections through the network.

- We introduce an optimization algorithm via parameter elimination that effectively and efficiently optimizes the linear coefficients and the neural network parameters .

- We systematically evaluate NIMO on synthetic and real datasets, verifying that it recovers correct feature effects, achieves accurate per-instance explanations, and offers a favorable trade-off between interpretability and predictive performance compared to existing methods.

## 2 RELATED WORK

**Interpretability in machine learning.** Interpretability refers to the capacity to express what a model has learned and the factors influencing its outputs in a manner that is clear and understandable to humans. In the literature, the two terms interpretability and explainability are sometimes used interchangeably. However, they have subtle yet important differences. Explainability focuses on providing reasons for the model's output. Most of the methods, especially in deep learning, provide post-hoc explanations. The most popular ones include LIME (Ribeiro et al., 2016), SHAP (Lundberg & Lee, 2017) and Grad-CAM (Selvaraju et al., 2020). On the other hand, interpretability focuses on understanding the inner mechanisms and decision processes of the model, and most of the methods that claimed to be interpretable involve inherently understandable models, such as decision trees (Wu et al., 2017; 2021), Lasso (Tibshirani, 1996) and Explainable Boosting Machines (Lou et al., 2013). The literature on interpretable machine learning is vast and we refer readers to Rudin et al. (2022) and Molnar (2025) for a more in-depth review.

**Linear Models and Input Feature Effects.** Coefficients from linear regression models provide inherently interpretable feature effects, directly quantifying how changes in input features influence predictions (Nguyen, 2020; Molnar, 2025). This transparency makes linear models especially attractive in domains where interpretability is critical. Regularization methods such as the Lasso (Tibshirani, 1996) further enhance interpretability by selecting the most relevant input features. However, linear models cannot capture nonlinearities or interactions without complicated feature engineering, e.g., basis expansion, which limits their ability to reflect complex data-generating processes. This motivates our approach, which preserves the interpretability of linear coefficients while extending their expressiveness to capture more complex data patterns.

**Hybrid models.** Several works have proposed combining linear models with neural networks to improve interpretability. Neural Additive Models (NAMs) (Agarwal et al., 2021) extend generalized additive models (GAMs) by learning one neural network per feature, whose outputs are summed to form the prediction. This design preserves interpretability at the feature level but does not have the ability to capture feature interactions. NODE-GAM (Chang et al., 2021) is also a neural generalized additive model that uses differentiable tree ensembles to learn interpretable univariate and pairwise shape functions. However, similar to NAM, both of their interpretability comes from the shape function, which is input dependent, and cannot provide a global population-level summary. LassoNet (Lemhadri et al., 2021) integrates a neural network with Lasso regression by enforcing that a feature can be used in the nonlinear part only if its corresponding linear coefficient is nonzero. Interpretable Mesomorphic Networks (IMN) (Kadra et al., 2024) use a hypernetwork to predict linear coefficients on a per-instance basis. While this increases flexibility, it sacrifices global interpretability of the coefficients, making it difficult to recover clear baseline effects. Interpretable Mixture of Experts (IME) (Ismail et al., 2022) routes each sample to an interpretable expert (e.g., a linear model), ensuring that for that sample the entire decision process is an exact and transparent explanation of which expert was chosen and why. While IME can maintain high accuracy and provide faithful local explanations, its global interpretability remains limited. In contrast, our proposed model (NIMO) preserves the global interpretability of linear coefficients while augmenting them with nonlinear corrections. This unifies global summaries via MEM with instance-level explanations within a single coherent framework.

## 3 PROPOSED APPROACH

### 3.1 A NONLINEAR INTERPRETABLE MODEL

**Model definition** Let $X \in \mathbb{R}^{n \times d}$ be $n$ observations with $d$ dimensions and let $\mathbf{y} \in \mathbb{R}^n$ be the targets. In the following, we assume the data $X$ to be standardized, i.e. that each feature has zero mean

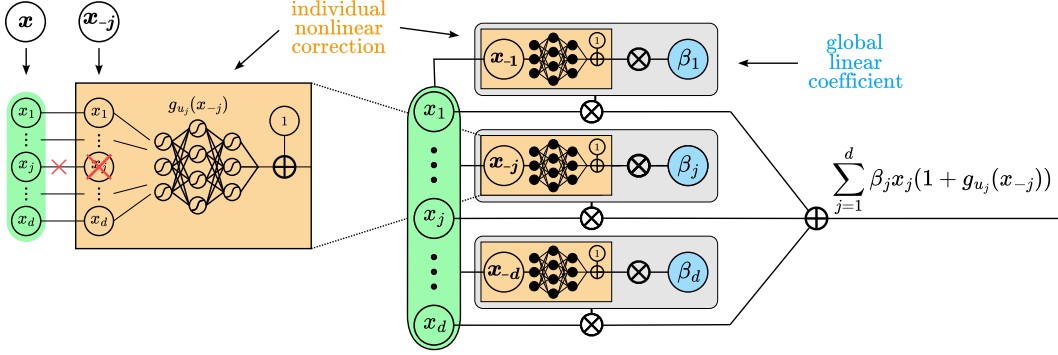

Figure 2: NIMO architecture. The model consists of global linear coefficients $\beta_j$ modulated by the individual data point $\mathbf{x}$ through $h_j$. In particular, $h_j$ depends on $\mathbf{x_{-j}}$, i.e. all features except the $j$-th.

and unit standard deviation. A linear regression model is simply defined as $y = \beta_0 + \sum_{j=1}^{d} x_j \beta_j$. In linear models the coefficient $\beta_j$ can be interpreted as the **additive effect** of the $j$-th feature on the output when all other feature values remain fixed. The idea of our model is to maintain the intelligible global interpretation **at the population level** while allowing for local explanations **at the per-instance level**. To do so, we define our approach starting from a linear model and multiplying the linear coefficients $\beta_j$ by a nonlinear correction term $h_j$ that depends on the data point $\mathbf{x}$:

$$f(\mathbf{x}) = \beta_0 + \sum_{j=1}^{d} x_j \beta_j \underbrace{(1 + g_{\boldsymbol{u}_j}(\mathbf{x_{-j}}))}_{h_j} \quad \text{s.t.} \quad g_{\boldsymbol{u}_j}(\mathbf{0}) = 0 , \tag{2}$$

where $\mathbf{x_{-j}}$ is the vector $\mathbf{x}$ without the $j$-th component and $g_{\boldsymbol{u}_j}(\cdot)$ are $d$ different scalar-valued functions defined by neural networks, each parametrized by $\boldsymbol{u}_j$. We provide an illustration of the proposed approach in Figure 2. Due to the specific design of our model, it exhibits the following properties. **First**, by explicitly excluding the $j$-th feature as input of the neural network, i.e. $g_{\boldsymbol{u}_j}(\mathbf{x_{-j}})$, $x_j$ contributes to the prediction only through the linear term $\beta_j$, preserving the behavior and interpretation of the coefficient $\beta_j$. **Second**, as we assume the data is standardized, the mean value of the features is zero. Therefore, since by construction $g_{\boldsymbol{u}_j}(\mathbf{0}) = 0$, when all other features are fixed at the mean value (i.e. at zero), the model reduces to a linear model. In other words, the marginal effect of feature $j$ is exactly $\beta_j$, as in a purely linear model. This ensures that the nonlinear correction term does not alter the global interpretability of the linear coefficients.

**Implementation details** A naive implementation of Eq. 2 requires $d$ separate neural networks, which is impractical in high-dimensional settings. To address this, we use a single shared network $g_{\boldsymbol{u}}$ and provide it with positional information to distinguish between features. Concretely, we append a positional encoding of the index $j$ to the masked input vector $\mathbf{x_{-j}}$, where the $j$-th component of $\mathbf{x}$ is set to zero. This way, $g_{\boldsymbol{u}}$ can learn feature-specific corrections while reusing parameters across all features. Finally, to enforce the interpretability constraint $g_{\boldsymbol{u}}(\mathbf{0}) = 0$, we simply subtract the mean prediction in each forward pass: $g_{\boldsymbol{u}}(\mathbf{x_{-j}}) \coloneqq g_{\boldsymbol{u}}(\mathbf{x_{-j}}) - g_{\boldsymbol{u}}(\mathbf{0})$.

### 3.2 Interpretability via marginal effects

Following Nguyen (2020), the **marginal effect (ME)** of feature $j$ at input $\mathbf{x}$ is defined as

$$\text{ME}_j = \frac{\partial f(\mathbf{x})}{\partial x_j}, \tag{3}$$

for any model $f(\mathbf{x})$. By definition, $\text{ME}_j$ measures the instantaneous change in prediction when varying $x_j$. However, marginal effects in nonlinear models are not constant, as they depend on the other input features. One way to summarize them is by computing **marginal effects at the mean (MEM)**, which estimates marginal effects at the average values of the input features:

$$\text{MEM}_j = \left. \frac{\partial f(\mathbf{x})}{\partial x_j} \right|_{\mathbf{x} = \bar{\mathbf{x}}} . \tag{4}$$

Based on the definition, we can compute the marginal effects for NIMO as below:

$$\text{ME}_j = \beta_j \big(1 + g_{\boldsymbol{u}_j}(\mathbf{x}_{-j})\big) + \sum_{i \neq j} x_i \beta_i \frac{\partial g_{\boldsymbol{u}_i}(\mathbf{x}_{-i})}{\partial x_j} \, ,$$

$$\text{MEM}_j = \beta_j \, .$$

Note that the mean value of input features $\bar{\mathbf{x}} = \mathbf{0}$ after standardization. Due to the design of our model, $\text{MEM}_j$ coincides with the linear coefficient. In principle, such marginal effects can also be computed via automatic differentiation for other differentiable models, but these do not necessarily yield interpretable results. In contrast, for NIMO, the MEMs directly correspond to the linear coefficients $\boldsymbol{\beta}$, capturing the additive population-level contribution of each feature.

While MEM is widely used as a global effect measure in statistics, it is important to acknowledge its general limitations, as discussed in Scholbeck et al. (2024). MEM is meaningful and interpretable in models that include an explicit global linear component, but for highly nonlinear functions the MEM may fail to reflect the true relevance of individual features. However, NIMO is explicitly designed to avoid such cases: its architecture explicitly separates a global linear component (captured by $\hat{\beta}$) from local, per-instance nonlinear adjustments. This structural constraint ensures that the global linear contribution remains identifiable and stable. Consequently, MEM remains appropriate and interpretable within NIMO's model class, even though it may be unsuitable as a universal feature-importance metric for fully unconstrained nonlinear functions.

### 3.3 Training via parameter elimination

For illustrative purposes, in this section we provide a high-level description of the optimization algorithm used to train NIMO. More detailed training procedure can be found in Appendix B.

**Optimization for sparse regression**   Consider the data $X \in \mathbb{R}^{n \times d}$ and targets $\mathbf{y} \in \mathbb{R}^n$. As argued before, we can model the $d$ networks with one network $g_{\boldsymbol{u}}$ and represent its outputs in a matrix $G_{\boldsymbol{u}} = g_{\boldsymbol{u}}(X) \in \mathbb{R}^{n \times d}$. The model in Eq. 2 can then be re-written in matrix form as $f(X) = B_{\boldsymbol{u}}\boldsymbol{\beta}$, where $B_{\boldsymbol{u}} = X + X \circ G_{\boldsymbol{u}}$ and "$\circ$" denotes element-wise multiplication. Let us first consider the standard ridge regression setting: $\mathcal{L}(\boldsymbol{\beta}, \boldsymbol{u}) = \|\mathbf{y} - B_{\boldsymbol{u}}\boldsymbol{\beta}\|^2 + \lambda\|\boldsymbol{\beta}\|^2$. To separate the optimization over $\boldsymbol{\beta}$ and $\boldsymbol{u}$, we take inspiration from the profile likelihood approach (Venzon & Moolgavkar, 1988; Murphy & Van der Vaart, 2000). The key idea is to eliminate $\boldsymbol{\beta}$ by first solving for its closed-form expression in terms of $\boldsymbol{u}$, and then substituting it back into the objective. This reduces the problem to an optimization over $\boldsymbol{u}$ only:

$$\min_{\boldsymbol{\beta}, \boldsymbol{u}} \mathcal{L}(\boldsymbol{\beta}, \boldsymbol{u}) \quad \longrightarrow \quad \min_{\boldsymbol{u}} \mathcal{L}(\hat{\boldsymbol{\beta}}(\boldsymbol{u}), \boldsymbol{u}), \quad \text{where} \quad \hat{\boldsymbol{\beta}}(\boldsymbol{u}) = (B_{\boldsymbol{u}}^T B_{\boldsymbol{u}} + \lambda I)^{-1} B_{\boldsymbol{u}}^T \mathbf{y}. \tag{5}$$

We can then efficiently optimize the objective with gradient descent over $\boldsymbol{u}$ only. Furthermore, within this framework we can incorporate sparsity by replacing the $\ell_2$ penalty with the $\ell_1$ penalty:

$$\min_{\boldsymbol{\beta}, \boldsymbol{u}} \|\mathbf{y} - B_{\boldsymbol{u}}\boldsymbol{\beta}\|^2 + \lambda\|\boldsymbol{\beta}\|_1 \, . \tag{6}$$

Unlike ridge regression, Lasso does not admit a closed-form solution for $\boldsymbol{\beta}$. To eliminate $\boldsymbol{\beta}$ in this setting, we use adaptive ridge regression, which admits at each step a closed-form expression for $\boldsymbol{\beta}$ in terms of $\boldsymbol{u}$ and is equivalent to Lasso at the optimum (Grandvalet, 1998); see Appendix A for a detailed proof. This enables us to efficiently optimize the problem using gradient descent while still enforcing sparsity. The full procedure is summarized in Algorithm 1 in the Appendix. A well-known limitation of Lasso is the over-shrinkage effect, where large coefficients are excessively penalized (Fan & Li, 2001). In some applications, this can lead to significant bias and degrade model performance. To mitigate this issue, we can extend the $\ell_1$ norm regularization to a sub-$\ell_1$ pseudo-norm, which penalizes large coefficients less aggressively while still promoting sparsity. A detailed introduction to this approach is provided in Appendix B.3.

**Extension to generalized linear models**   The same idea can be extended to any generalized linear model of the form: $f(\mathbb{E}[Y|\mathbf{x}]) = \beta_0 + \beta_1 x_1 + \cdots + \beta_p x_p = \mathbf{x}^T \boldsymbol{\beta}$, where $f$ is a link function. For simplicity, consider logistic regression with $\ell_1$ penalty. As before, we first get a closed-form expression for $\boldsymbol{\beta}$ in terms of $\boldsymbol{u}$ and then substitute it back to optimize over the neural network

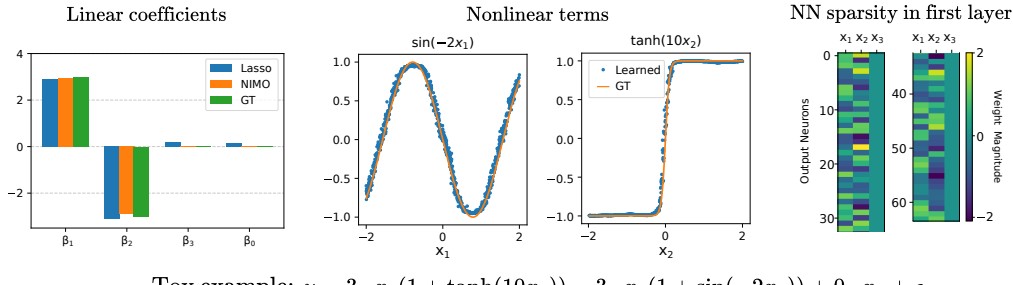

Toy example: $y = 3 \cdot x_1(1 + \tanh(10x_2)) - 3 \cdot x_2(1 + \sin(-2x_1)) + 0 \cdot x_3 + \epsilon$

Figure 3: Illustrative toy example. *Left*: NIMO learns the GT coefficients and also finds the correct sparsity. *Middle*: the learned nonlinearities coincide with the GT. *Right*: the first `fc` layer of the NN learns the correct sparsity: only features $x_1$ and $x_2$ have non-zero weights, while $x_3$ is ignored.

parameters $\boldsymbol{u}$. What is different is that in each optimization step we use the iteratively reweighted least squares (IRLS) surrogate (Green, 1984; McCullagh & Nelder, 1989), which is a weighted least-squares approximation of the original problem:

$$\min_{\boldsymbol{\beta}, \boldsymbol{u}} \; \tfrac{1}{2}\big\|W^{1/2}(\mathbf{z} - B_{\boldsymbol{u}}\boldsymbol{\beta})\big\|^2 + \lambda\|\boldsymbol{\beta}\|_1 \;, \tag{7}$$

where $W_{ii} = \sigma(\eta_i)\big[1 - \sigma(\eta_i)\big]$ and $\boldsymbol{\eta} = B_{\boldsymbol{u}}\boldsymbol{\beta}$. The so-called working response $\mathbf{z}$ is computed as $\mathbf{z} = \boldsymbol{\eta} + W^{-1}\big(\mathbf{y} - \sigma(\boldsymbol{\eta})\big)$. Eq. 7 is also a Lasso problem and we can employ adaptive ridge regression to find a closed form expression for $\hat{\boldsymbol{\beta}}(\boldsymbol{u})$. Then, we substitute $\hat{\boldsymbol{\beta}}(\boldsymbol{u})$ back and optimize over the neural networks parameters $\boldsymbol{u}$ via gradient descent. Appendix B.2 includes a detailed derivation.

## 3.4 PROOF-OF-CONCEPT: A TOY EXAMPLE

We illustrate NIMO on a simple three-dimensional toy example. The data is generated as:

$$y = 3 \cdot x_1 \cdot (1 + \tanh(10x_2)) + (-3) \cdot x_2 \cdot (1 + \sin(-2x_1)) + 0 \cdot x_3 + \epsilon \;, \tag{8}$$

where $\epsilon \sim \mathcal{N}(0, 0.1^2)$ is Gaussian noise. This example involves three features $\{x_1, x_2, x_3\}$, but $x_3$ is inactive since it is multiplied by a zero coefficient. We generate 400 samples and use a 200-100-100 train-validation-test split. As a baseline, we compare against Lasso regression. We further apply group $\ell_2$ regularization to the weight matrix of the first fully connected layer in the neural network to enhance interpretability. Further implementation details are given in Appendix C.2. The results are shown in Figure 3. On the left, we see that the estimated coefficients $\boldsymbol{\beta}$, which also represent the marginal effects at the mean, learned by NIMO align with the ground truth; in particular, NIMO correctly identifies $\beta_3 = 0$, i.e. that $x_3$ is uninformative. In the middle, the nonlinear corrections $g_j(\mathbf{x}_{-j})$ recover the true underlying interactions for both $x_1$ and $x_2$. This highlights an important interpretability aspect: while $\beta_1 = 3$ and $\beta_2 = -3$ represent the global population-level effects, the neural network outputs vary across individual inputs, providing per-instance corrections. Therefore, NIMO not only preserves the global interpretability of the coefficients but also captures the correct local deviations at the per-instance level. Finally, the right panel shows that neurons connected to $x_3$ in the first fully connected layer are sparse, permitting an additional layer of interpretability: $x_3$ does not contribute to nonlinear interactions.

## 4 EXPERIMENTS

In this section, we experimentally evaluate the proposed NIMO, focusing on interpretability through marginal effects. We first introduce the baseline methods and the experimental protocol used for comparison. Next, we formulate three hypotheses and present empirical results that examine both the interpretability and predictive performance of NIMO. Additional results are provided in Appendix D.

### 4.1 METHODS AND PROTOCOL

**Methods:** In this paper, we compare our method against **Lasso** (Tibshirani, 1996), a vanilla neural network (**NN**) (Goodfellow et al., 2016), and several state of the art interpretable ap-

proaches, namely **LassoNet** (Lemhadri et al., 2021), Neural Additive Models (**NAMs**) (Agarwal et al., 2021) and Interpretable Mesomorphic Networks (**IMNs**) (Kadra et al., 2024).

LassoNet encourages the network to use only a subset of the available input features by enforcing sparsity with a novel objective. NAMs learn a linear combination of neural networks that each attend to a single input feature. IMNs optimize deep hypernetworks to generate explainable linear models on a per-instance basis.

**Protocol:** We conduct experiments on both synthetic and real datasets to verify the interpretability and performance of our model. For synthetic datasets, we create them by explicitly controlling the linear coefficients and the nonlinearities, similarly to Eq. 8. We explore multiple

Table 1: MSE loss for synthetic regression settings.

| Method (#features) | Setting 1 (5) | Setting 2 (10) | Setting 3 (50) |
|---|---|---|---|
| Lasso | 3.164 | 3.340 | 13.122 |
| NN | 1.109 | 1.482 | 13.718 |
| NAM | 3.427 | 5.126 | 16.543 |
| IMN | 0.137 | 1.188 | 6.308 |
| LassoNet | 0.078 | 2.612 | 1.738 |
| NIMO | **0.024** | **0.197** | **0.380** |

nonlinearities, different dimensionalities and also use several uninformative features (i.e. with zero coefficient) to test if we can recover sparse signals. The details about the exact data generating process are provided in the Appendix D.2. For real datasets, we choose several popular and well-studied datasets from UCI Machine Learning Repository [1].

## 4.2 HYPOTHESES AND EXPERIMENT RESULTS

**Hypothesis 1:** NIMO outperforms other methods on synthetic datasets and remains robust in low-data regime.

We compare NIMO with the above methods on the synthetic datasets and evaluate predictive performance using mean squared error (MSE). The results are presented in Table 1. In each setting, we use 200 samples for training. In this low-data regime, the naive neural network tends to overfit even with a relatively shallow MLP, while the linear model is too simple to capture the complex underlying nonlinearities. NAM, which assigns a separate MLP to each feature component, is even more prone to overfitting than the naive neural network. Overall, the results show that NIMO not only outperforms the other methods by effectively capturing complex nonlinear feature interactions, but also remains robust in low-data settings due to the regularization and optimization strategies employed.

**Hypothesis 2:** NIMO accurately estimates the marginal effects at the mean (MEM), preserving the additive contribution of each feature at the population-level.

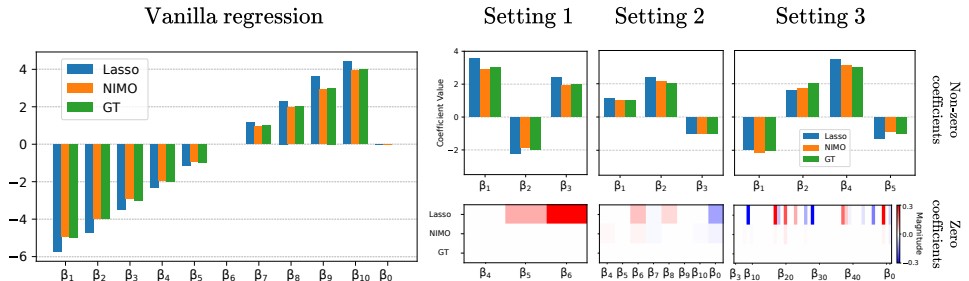

Figure 4: Learned $\beta$ coefficients on synthetic regression datasets and comparison with Lasso regression and Ground truth. Vanilla regression (*left*) is purely linear while Settings 1, 2, 3 (*right*) have different nonlinearities. Both NIMO and Lasso recover the correct nonzero coefficients (informative features) but only NIMO recovers the correct sparsity of the zero coefficients (uninformative features).

This experiment evaluates whether NIMO can recover population-level interpretability in terms of marginal effects at the mean. For the other methods that rely on automatic differentiation, MEM can certainly be computed, but they do not yield meaningful interpretations. They reduce to arbitrary

---

[1]https://archive.ics.uci.edu/datasets

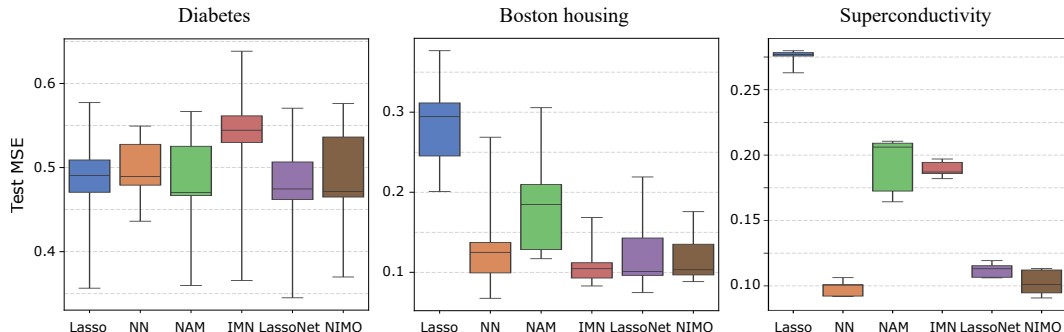

Figure 5: Predictive performance (MSE) the diabetes, Boston housing and superconductivity dataset.

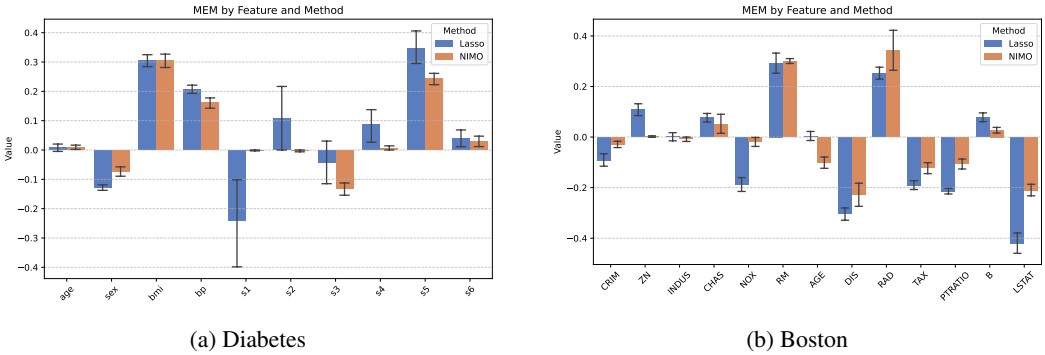

(a) Diabetes

(b) Boston

Figure 6: MEMs for Lasso and NIMO on the diabetes (*left*) and Boston housing dataset (*right*).

numerical values that neither explain the prediction for an instance nor capture population-level effects. In contrast, the specific structure of NIMO ensures that the marginal effects at the mean coincide with the linear coefficients, as in the linear models, providing a clear and intelligible interpretation. Therefore, we restrict the comparison to Lasso only, which directly yields globally interpretable coefficients. As a sanity check, we first test NIMO on a *Vanilla regression* dataset with purely linear coefficients in $\{-5, -4, \ldots, 3, 4\}$ and no nonlinearities. This verifies whether the neural network component of NIMO interferes with the linear part when no nonlinear effects are present. As shown in Figure 4 (*left*), NIMO perfectly recovers the ground truth coefficients, confirming that the neural network does not distort the linear coefficients, maintaining the intelligibility of MEM. We then extend the analysis to three additional regression settings (*Setting 1, 2, 3*) with varying nonlinearities and sparsity levels (see Appendix D.2 for details). Unlike Lasso, NIMO not only recovers the true coefficients but also correctly identifies the sparsity pattern, i.e., which features are uninformative (Figure 4, *bottom-right*). Based on these results, we conclude that NIMO estimates the accurate marginal effects at the mean and preserves the additive population-level contribution of each feature.

**Hypothesis 3:** NIMO achieves a favorable trade-off between interpretability and predictive performance in practice.

This experiment aims to evaluate whether NIMO can achieve strong predictive performance on real datasets and how the results can be interpreted. We select several widely used datasets from the UCI Machine Learning Repository: the diabetes dataset (Efron et al., 2004), the Boston housing dataset (Belsley et al., 2005), and the superconductivity dataset (Hamidieh, 2018). Each dataset is randomly shuffled and split multiple times with different seeds (9 seeds for the diabetes dataset and the Boston housing dataset, 5 seeds for the superconductivity dataset), and the experiments are performed once on each split using identical model settings. We first examine the predictive performance of NIMO. The box plot of the mean squared error for each method is shown in Figure 5. NIMO generally achieves performance compatible with the other methods and attains the best performance on the superconductivity dataset. To assess interpretability, we first visualize the marginal effects at the mean (MEM) obtained from Lasso and NIMO for the diabetes and Boston housing datasets

in Figure 6. Recall that MEM estimates marginal effects at the average values of the input features, reflecting the additive population-level contribution of each feature. We observe that the MEM profiles of Lasso and NIMO differ across several features. These differences do not necessarily imply the presence of nonlinear feature interactions; rather, both datasets contain groups of strongly correlated predictors, making feature selection inherently unstable. Consequently, Lasso and NIMO may highlight different features within the same correlated group, even if their predictive roles are similar. In Appendix D.8, we experimentally showcase the degree of nonlinearities in each dataset. For the superconductivity dataset, which involves a much larger set of features, the corresponding MEM visualization is provided in the Appendix in Figure 14.

We then use MEM to rank the features on the diabetes and Boston housing datasets. For comparison, we also apply the post-hoc explainer SHAP (Lundberg & Lee, 2017) to our model and rank features according to the global SHAP values. The results are shown in Table 2. On the diabetes dataset, all methods behave similarly in identifying the most important features, which is expected since the data lacks complex nonlinear interactions. This is further supported by the comparable predictive performance across methods, as shown in Figure 5 (*left*). Notably, SHAP produces identical feature rankings to MEM for NIMO. In contrast, for the Boston housing dataset, the methods yield inconsistent feature rankings, suggesting that nonlinear interactions play a crucial role in explaining the data. Although MEM can also be computed for other nonlinear methods, they do not provide the same level of intelligible interpretability as our model. For NIMO, SHAP rankings are largely consistent with MEM, though not perfectly aligned. The most pronounced difference appears with LassoNet, which assigns high importance to features considered least relevant by other methods, such as INDUS. We attribute this to how LassoNet handles nonlinearities.

Table 2: Feature rankings according to MEMs on the diabetes and Boston housing datasets. A lower ranking is associated with a higher feature importance. Features whose rankings are consistently similar across multiple methods are highlighted in yellow.

Diabetes Dataset

| Feature | SHAP | NIMO | LassoNet | NAM | IMN | Lasso |
|---|---|---|---|---|---|---|
| age | 7 | 7 | 9 | 5 | 9 | 10 |
| sex | 5 | 5 | 5 | 7 | 5 | 5 |
| bmi | 1 | 1 | 2 | 2 | 2 | 2 |
| bp | 3 | 3 | 3 | 3 | 3 | 4 |
| s1 | 10 | 10 | 7 | 10 | 8 | 3 |
| s2 | 9 | 9 | 10 | 9 | 7 | 6 |
| s3 | 4 | 4 | 4 | 6 | 6 | 8 |
| s4 | 8 | 8 | 8 | 8 | 4 | 7 |
| s5 | 2 | 2 | 1 | 1 | 1 | 1 |
| s6 | 6 | 6 | 6 | 4 | 10 | 9 |

Boston Housing Dataset

| Feature | SHAP | NIMO | LassoNet | NAM | IMN | Lasso |
|---|---|---|---|---|---|---|
| CRIM | 10 | 9 | 3 | 5 | 8 | 9 |
| ZN | 13 | 13 | 10 | 11 | 12 | 8 |
| INDUS | 11 | 12 | 2 | 12 | 13 | 13 |
| CHAS | 12 | 8 | 5 | 8 | 10 | 11 |
| NOX | 8 | 11 | 9 | 10 | 9 | 7 |
| RM | 2 | 2 | 8 | 6 | 1 | 3 |
| AGE | 6 | 7 | 6 | 9 | 5 | 12 |
| DIS | 3 | 3 | 1 | 3 | 3 | 2 |
| RAD | 5 | 1 | 12 | 13 | 7 | 4 |
| TAX | 4 | 5 | 11 | 7 | 6 | 6 |
| PTRATIO | 7 | 6 | 4 | 4 | 4 | 5 |
| B | 9 | 10 | 13 | 2 | 11 | 10 |
| LSTAT | 1 | 4 | 7 | 1 | 2 | 1 |

## 4.3 ABLATION STUDY

To verify the effectiveness of masking out the $j$-th component of an instance when passing it through the neural network, we conducted an ablation study. Specifically, we examined the effect of masking out the $j$-th component as well as the effect of removing position encoding. From the ablation results across the three settings (Figure 7, Figure 15, and Figure 16), we observe several consistent patterns:

- Without masking the $j$-th component, the first `fc` layer of the neural network can still exhibit sparsity patterns, but they are less distinct than when masking is applied. We attribute this to the position encoding, which enables the network to identify the feature index.

- Without masking the $j$-th component, the MEMs are confounded by nonlinear interactions, so their interpretation as additive regression coefficients and the meaning of their magnitudes are lost.

- Further removing the position encoding distorts the sparsity patterns in the first `fc` layer and alters the magnitude of the coefficients, undermining interpretability.

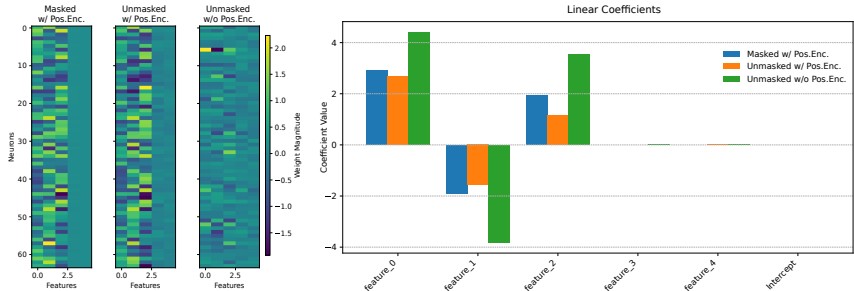

Figure 7: The effect of feature masking and position encoding on the linear coefficients and the sparsity of the first `fc` layer on setting 1.

## 4.4 LIMITATIONS

The proposed approach is designed to preserve the global, population-level interpretability inherited from linear models while simultaneously learning nonlinear corrections to linear predictions through neural networks, thereby enhancing the expressive capacity of linear models. Thus, the underlying assumption is that the data can be well described by adding nonlinear refinements on top of a linear model. In settings dominated by strong self-interactions or highly nonlinear regimes, NIMO cannot fully capture the underlying relationships in an interpretable manner. Assume the data-generating process contains a self-interacting term of the form $x_j f(x_j)$, where $f$ is a linear or nonlinear function (e.g. $x \sin(x)$ or $x^2$). To maintain interpretability of the linear coefficients, we imposed the constraint that $g_{\boldsymbol{u}_j}(\mathbf{x}_{-\mathbf{j}})$ does not depend on $x_j$, which prevents NIMO from learning such self-interacting nonlinearities.

However, if such a feature is known to play an important role in the dataset, simple feature engineering can resolve the issue. To experimentally and explicitly illustrate both the limitation of NIMO and a straightforward remedy, we conduct an ablation study on a synthetic dataset. The data are generated as

$$y = x_1^2 + x_2^2 + x_3^2 + \epsilon. \tag{9}$$

Empirically, we demonstrate that naively applying NIMO to this dataset results in poor performance. However, after applying a polynomial basis expansion of degree 2, NIMO can perfectly solve the problem while still providing global, population-level interpretability through the linear coefficients $\boldsymbol{\beta}$. More detailed experimental results can be found in Appendix D.6.

## 5 CONCLUSIONS

In this paper, we propose a nonlinear interpretable model (NIMO). Unlike other hybrid approaches, our model preserves the interpretability of linear coefficients, capturing the additive contribution of each feature to the prediction, while also accommodating nonlinear interactions. Interpretability is formalized through marginal effects, with the marginal effects at the mean (MEM) providing global, population-level summaries that align directly with the linear coefficients. To enable effective training, we introduce an optimization algorithm based on parameter elimination, and enforce sparsity through adaptive ridge regularization. We demonstrate the effectiveness of the proposed approach on both synthetic and real datasets, showing competitive predictive performance while delivering meaningful and globally consistent interpretations.

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

APPENDIX

## A    EQUIVALENCE OF ADAPTIVE RIDGE REGRESSION AND LASSO

The equivalence in the solution of Adaptive ridge regression and Lasso regression is a well-known fact (Tibshirani, 1996; Grandvalet, 1998; Grandvalet & Canu, 1998). Since both our model relies on this fact both in the regression and classification settings, we report its proof below.

*Proof.* Let $X \in \mathbb{R}^{n \times d}$ be $n$ $d$-dimensional observations and $\mathbf{y} \in \mathbb{R}^n$ be the targets. In a linear model the relationship between target and data is assumed to be linear:

$$\mathbf{y} = X\boldsymbol{\beta} + \epsilon \,, \tag{10}$$

where $\boldsymbol{\beta} \in \mathbb{R}^d$ is the coefficient vector and $\epsilon \sim \mathcal{N}(0,1)$ is standard Gaussian noise. In Ridge Regression a penalized version of the objective is minimized:

$$\min_{\boldsymbol{\beta}} \|X\boldsymbol{\beta} - \mathbf{y}\|_2^2 + \lambda \sum_{j=1}^d \beta_j^2 \,, \tag{11}$$

where $\lambda$ weighs the penalty term. In Adaptive Ridge regression, instead of a single $\lambda$, a set of feature-specific penalties $\{\nu_i\}_{i=1}^d$ are introduced. The objective then reads as

$$\min_{\boldsymbol{\beta}\,\boldsymbol{\nu}} \|X\boldsymbol{\beta} - \mathbf{y}\|^2 + \sum_{i=1}^d \nu_i \beta_i^2 \quad \text{s. t.} \quad \sum_{i=1}^d \frac{1}{\nu_i} = \frac{d}{\lambda} \,, \tag{12}$$

where $\nu_j \geq 0 \; \forall j$ and $\lambda$ is a predefined value. One can solve the above constrained optimization through the method of Lagrange multipliers:

$$\mathcal{L}(\boldsymbol{\nu}, \mu) = \|X\boldsymbol{\beta} - \mathbf{y}\|^2 + \sum_{i=1}^d \nu_i \beta_i^2 + \mu \sum_{i=1}^d \frac{1}{\nu_i} \,, \tag{13}$$

where $\mu$ is the Lagrangian multiplier and $\boldsymbol{\nu} := [\nu_1, \ldots, \nu_d]^T$. The stationary points can be found by taking the gradients with respect to $\nu_k$, as follows:

$$\frac{\partial \mathcal{L}(\boldsymbol{\nu}, \mu)}{\partial \nu_k} = \beta_k^2 - \mu \nu_k^{-2} = 0 \quad \Rightarrow \quad \nu_k \beta_k^2 = \mu/\nu_k \quad \text{or} \quad \nu_k |\beta_k| = \mu^{1/2} \tag{14}$$

If we now substitute $\nu_k |\beta_k| = \mu^{1/2}$ in the Adaptive Ridge objective in Eq. 12, we get the Lasso objective:

$$
\begin{aligned}
f(\boldsymbol{\beta}, \boldsymbol{\nu}) &= \|X\boldsymbol{\beta} - \mathbf{y}\|^2 + \sum_{i=1}^d \nu_i \beta_i^2 \\
&= \|X\boldsymbol{\beta} - \mathbf{y}\|^2 + \sum_{i=1}^d \nu_i |\beta_i| \cdot |\beta_i| \\
&= \|X\boldsymbol{\beta} - \mathbf{y}\|^2 + \sum_{i=1}^d \mu^{1/2} |\beta_i| \\
&= \|X\boldsymbol{\beta} - \mathbf{y}\|^2 + \mu^{1/2} \sum_{i=1}^d |\beta_i|.
\end{aligned}
\tag{15}
$$

which gives the relationship between adaptive ridge regression and Lasso regression, with the Lasso penalty parameter at optimum to be $\mu^{1/2} = \frac{\lambda}{d} \sum_k |\beta_k|$:

$$
\begin{aligned}
\nu_k \beta_k^2 = \mu/\nu_k \quad &\Rightarrow \quad \sum_k \nu_k \beta_k^2 = \sum_k \frac{\mu}{\nu_k} \overset{equation\ 12}{=} \frac{\mu d}{\lambda} \\
&\Rightarrow \quad \mu = \frac{\lambda}{d} \sum_k \nu_k \beta_k^2 = \frac{\lambda}{d} \sum_k \nu_k |\beta_k| \cdot |\beta_k| \overset{equation\ 14}{=} \frac{\lambda}{d} \sum_k |\beta_k| \mu^{1/2} \\
&\Rightarrow \quad \mu^{1/2} = \frac{\lambda}{d} \sum_k |\beta_k|
\end{aligned}
\tag{16}
$$

$\square$

# B TRAINING VIA PARAMETER ELIMINATION

The proposed model in Eq. 2 has two sets of parameters: the linear coefficients $\boldsymbol{\beta}$ and the neural network parameters $\boldsymbol{u}$. Since these two sets of parameters are tightly entangled, jointly optimizing them via gradient descent is not straightforward in practice. Furthermore, to increase the interpretability of the model, we also aim to achieve sparsity in the learned coefficients. In this section, we show how to achieve these optimization goals via parameter elimination. Specifically, we eliminate $\boldsymbol{\beta}$ by expressing it in terms of the neural network parameters $\boldsymbol{u}$ and substituting it back into the optimization objective. The resulting problem can then be optimized effectively and efficiently using gradient descent. Sparsity is achieved by reformulating the problem as an adaptive ridge regression. We further show that this approach can be extended to generalized linear models, and we illustrate it in detail for logistic regression.

## B.1 REGRESSION SETTING: ADAPTIVE RIDGE REGRESSION

We already showed that the proposed model in Eq. 2 can be re-written in matrix form as:

$$f(X) = B_{\boldsymbol{u}}\boldsymbol{\beta} \quad \text{with} \quad B_{\boldsymbol{u}} = X + X \circ G_{\boldsymbol{u}} \tag{17}$$

where $\boldsymbol{\beta} \in \mathbb{R}^d$ are the linear coefficients, $G_{\boldsymbol{u}} \in \mathbb{R}^{n \times d}$ is the neural network output with inputs $X \in \mathbb{R}^{n \times d}$ and "$\circ$" denotes element-wise multiplication. The linear model part is then expressed as:

$$\mathbf{y} = B_{\boldsymbol{u}}\boldsymbol{\beta} + \epsilon, \tag{18}$$

where $\mathbf{y} \in \mathbb{R}^n$ denotes the targets. Adaptive Ridge regression minimizes the following objective:

$$\min_{\boldsymbol{\beta}, \boldsymbol{u}} \|B_{\boldsymbol{u}}\boldsymbol{\beta} - \mathbf{y}\|^2 + \sum_{i=1}^{d} \nu_i \beta_i^2 \quad \text{s. t.} \quad \sum_{i=1}^{d} \frac{1}{\nu_i} = \frac{d}{\lambda} \tag{19}$$

where $\nu_i > 0$ are the penalty parameters for each coefficient $\beta_i$, and $\lambda > 0$ is a predefined parameter. To avoid divergent solutions we use the re-parameterization employed in Grandvalet (1998); Grandvalet & Canu (1998):

$$\gamma_i = (\nu_i/\lambda)^{1/2} \beta_i, \quad c_i = (\lambda/\nu_i)^{1/2} . \tag{20}$$

If we substitute $\beta_i$ and $\nu_i$ into the Adaptive Ridge regression objective, we get the following optimization problem:

$$\min_{\boldsymbol{\gamma}, \boldsymbol{c}, \boldsymbol{u}} \|\mathbf{y} - B_{\boldsymbol{u}} D_{\boldsymbol{c}} \boldsymbol{\gamma}\|^2 + \lambda \|\boldsymbol{\gamma}\|^2 \quad \text{s. t.} \quad \sum_{i=1}^{d} c_i^2 = d, \quad c_i \geq 0, \tag{21}$$

where $D_{\boldsymbol{c}}$ is a diagonal matrix whose diagonal elements are given by the vector $\boldsymbol{c} := [c_1, \ldots, c_d]^T$. Applying the parameter elimination trick, if we fix $\boldsymbol{c}$ and $\boldsymbol{u}$, $\boldsymbol{\gamma}$ can be expressed in a closed-form in terms of $\boldsymbol{c}$ and $\boldsymbol{u}$:

$$\hat{\boldsymbol{\gamma}}(\boldsymbol{c}, \boldsymbol{u}) = \left(D_{\boldsymbol{c}} B_{\boldsymbol{u}}^T B_{\boldsymbol{u}} D_{\boldsymbol{c}} + \lambda I\right)^{-1} D_{\boldsymbol{c}} B_{\boldsymbol{u}}^T \mathbf{y}, \tag{22}$$

We can now substitute $\hat{\boldsymbol{\gamma}}(\boldsymbol{c}, \boldsymbol{u})$ back into the original problem and take the constraint into consideration. This leads us to a Lagrangian minimization problem in the scaling variable $\boldsymbol{c}$ and neural network parameters $\boldsymbol{u}$:

$$\min_{\boldsymbol{c}, \boldsymbol{u}} \|\mathbf{y} - B_{\boldsymbol{u}} D_{\boldsymbol{c}} \hat{\boldsymbol{\gamma}}(\boldsymbol{c}, \boldsymbol{u})\|^2 + \lambda \|\hat{\boldsymbol{\gamma}}(\boldsymbol{c}, \boldsymbol{u})\|^2 + \mu \|\boldsymbol{c}\|^2, \tag{23}$$

where $\mu$ is the Lagrangian multiplier introduced for $\boldsymbol{c}$, and we can optimize $\boldsymbol{c}, \boldsymbol{u}$ together through gradient descent. The training algorithm for this re-parameterized adaptive ridge regression is summarized in Alg. 1.

---

**Algorithm 1** Training of Re-parameterized Adaptive Ridge Regression

---

**Require:** Training data $X \in \mathbb{R}^{n \times d}$, training target $\mathbf{y} \in \mathbb{R}^n$, neural network $g_{\boldsymbol{u}}(\cdot)$, regularization
   parameter $\lambda$ and $\mu$, learning rate $\eta$, maximum iterations $T$.
1: Initialize network parameters $\boldsymbol{u}$ and scaling coefficients $\boldsymbol{c}$ (with $c_i \geq 0$).
2: **for** $t = 1, \ldots, T$ **do**
3:    $B_{\boldsymbol{u}} = X + X \circ G_{\boldsymbol{u}}$              // Compute modified design matrix
4:    $D_{\boldsymbol{c}} = \mathrm{diag}(\boldsymbol{c})$               // Form diagonal scaling matrix
5:    $\hat{\boldsymbol{\gamma}} = \left(D_{\boldsymbol{c}} B_{\boldsymbol{u}}^T B_{\boldsymbol{u}} D_{\boldsymbol{c}} + \lambda I\right)^{-1} D_{\boldsymbol{c}} B_{\boldsymbol{u}}^T \mathbf{y}$                  // Solve for $\hat{\boldsymbol{\gamma}}$
6:    $\hat{\boldsymbol{\beta}} = \boldsymbol{c} \circ \hat{\boldsymbol{\gamma}}$           // Compute regression coefficients
7:    $L = \|\mathbf{y} - B_{\boldsymbol{u}} \hat{\boldsymbol{\beta}}\|^2 + \lambda \|\hat{\boldsymbol{\gamma}}\| + \mu \|\boldsymbol{c}\|^2$              // Compute loss function
8:    $\boldsymbol{c} \leftarrow \boldsymbol{c} - \eta \nabla_{\boldsymbol{c}} L, \quad \boldsymbol{u} \leftarrow \boldsymbol{u} - \eta \nabla_{\boldsymbol{u}} L$      // Gradient descent updates
9: **end for**
10: **Return** $\hat{\boldsymbol{\beta}}, \boldsymbol{c}, \boldsymbol{u}$                  // Output final parameters

---

### B.2  LOGISTIC REGRESSION: ITERATIVELY REWEIGHTED LEAST SQUARES

The procedure followed in the previous section can be extended to generalized linear models. Here
we showcase how to do it for logistic regression. Logistic regression is a classification model that
describes the probability that a binary outcome $y \in \{0, 1\}$ occurs given a feature vector $\mathbf{x} \in \mathbb{R}^d$:

$$P(y = 1|\mathbf{x}) = \sigma(\mathbf{x}^T \boldsymbol{\beta}) \tag{24}$$

where $\sigma(z) = \frac{1}{1+e^{-z}}$ is the sigmoid function. The model can then be trained by minimizing the
negative log-likelihood. Similarly to the regression case discussed in the previous section, we add an
adaptive ridge regression penalty to the objective:

$$\mathcal{L}(\boldsymbol{\beta}) = -\sum_{i=1}^n [y_i \log \sigma(\mathbf{x}_i^T \boldsymbol{\beta}) + (1 - y_i) \log(1 - \sigma(\mathbf{x}_i^T \boldsymbol{\beta}))] + \sum_{j=1}^d \nu_j \beta_j^2 \quad \text{s. t.} \quad \sum_{j=1}^d \frac{1}{\nu_j} = \frac{d}{\lambda} \tag{25}$$

Now we can use the same re-parameterization used in the previous section for regression. We get the
following standard ridge-penalized logistic regression:

$$\min_{\boldsymbol{\gamma}, \boldsymbol{c}} -\sum_{i=1}^n \left[ y_i \log \sigma \left( (\mathbf{x}_i \odot \boldsymbol{c})^T \boldsymbol{\gamma} \right) + (1 - y_i) \log \left( 1 - \sigma \left( (\mathbf{x}_i \odot \boldsymbol{c})^T \boldsymbol{\gamma} \right) \right) \right] + \lambda \|\boldsymbol{\gamma}\|^2$$

$$\text{s.t.} \quad \sum_{j=1}^d c_j^2 = d, \quad c_j \geq 0, \tag{26}$$

where $\boldsymbol{c} := [c_1, ..., c_d]^T$, $\odot$ is element-wise multiplication. The same as what we done for regression,
we can eliminate $\boldsymbol{\gamma}$ by expressing it in a closed-form with respect to $\boldsymbol{c}$. To get this closed-form
expression of $\boldsymbol{\gamma}$, we use the iteratively reweighted least squares (IRLS) algorithm. The associated
update reads as:

$$\boldsymbol{\gamma}^{(t+1)}(\boldsymbol{c}) = (\tilde{X}^T W^{(t)} \tilde{X} + \lambda I)^{-1} \tilde{X}^T W^{(t)} \mathbf{z}^{(t)} \tag{27}$$

where $\tilde{X}$ is the modified design matrix with entries $\tilde{x}_{ij} = c_j x_{ij}$, $W^{(t)}$ is a diagonal weight matrix
with entries $w_i^{(t)} = p_i^{(t)}(1 - p_i^{(t)})$, $p_i^{(t)} = \frac{1}{1+e^{-\tilde{\boldsymbol{x}}_i^T \boldsymbol{\gamma}^{(t)}}}$, and $\mathbf{z}^{(t)}$ is the the vector of working response
with entries $z_i^{(t)} = \tilde{\boldsymbol{x}}_i^T \boldsymbol{\gamma}^{(t)} + \frac{y_i - p_i^{(t)}}{w_i^{(t)}}$. We then substitute $\boldsymbol{\gamma}^{(t+1)}(\boldsymbol{c})$ back into the original problem,
resulting the optimization over $\boldsymbol{c}$ only. Note that in our model, our design matrix $X$ contains the
nonlinear term as well, so we would need to replace $X$ with $B_{\boldsymbol{u}} = X + X \circ G_{\boldsymbol{u}}$. The rest goes the
same as the case for regression.

### B.3 EXTENSION TO SUB-$\ell_1$ PSEUDO-NORMS

In some cases sparsity is a crucial aspect in the learning process. One of the limitations of lasso is the well-known over-shrinkage effect, which means that shrinkage of uninformative features could be achieved at the expense of shrinking the retained features as well. One way to avoid this is to use sub-$\ell_1$ pseudo-norms, which in the limits of the $\ell_0$ pseudo-norm achieve sparsity without any shrinkage. The main challenge with sub-$\ell_1$ pseudo-norm is that the resulting objective function becomes non-convex and harder to optimize. However, recent work has successfully used sub-$\ell_1$ norms as a prior on regression problems (Negri et al., 2023), which is similar to our setting.

We now show how to use a sub-$\ell_1$ prior in our Adaptive Ridge regression objective, which we report below for completeness:

$$\min_{\boldsymbol{\beta}} \|X\boldsymbol{\beta} - \mathbf{y}\|^2 + \sum_{i=1}^{d} \nu_i \beta_i^2 \quad \text{s.t.} \quad \sum_{i=1}^{d} \frac{1}{\nu_i} = \frac{d}{\lambda} \tag{28}$$

In order to generalize to sub-$\ell_1$ pseudo-norms, we need the following simple modification on $\nu_i$:

$$\sum_{i=1}^{d} \frac{1}{\nu_i^{\delta}} = C, \quad 0 < \delta \leq 1. \tag{29}$$

We now apply the same re-parameterization:

$$\gamma_i = (\nu_i/\lambda)^{1/2} \beta_i, \quad c_i = (\lambda/\nu_i)^{1/2} . \tag{30}$$

The objective function then reads as

$$\min_{\boldsymbol{\gamma}, \boldsymbol{c}} \|\mathbf{y} - XD_{\boldsymbol{c}}\boldsymbol{\gamma}\|^2 + \lambda\|\boldsymbol{\gamma}\|^2 \quad \text{s.t.} \quad \sum_{i=1}^{d} c_i^{2\delta} = \lambda^{\delta} C, \quad c_i > 0 . \tag{31}$$

Fixing $\boldsymbol{c}$, we can eliminate $\boldsymbol{\gamma}$ by expressing it in a closed-form in terms of $\boldsymbol{c}$:

$$\hat{\boldsymbol{\gamma}}(\boldsymbol{c}) = \left(D_{\boldsymbol{c}} X^T X D_{\boldsymbol{c}} + \lambda I\right)^{-1} D_{\boldsymbol{c}} X^T \mathbf{y}. \tag{32}$$

Substituting it back into the original optimization problem and taking the constraint into consideration lead us to a Lagrangian minimization problem in the scaling variable $\boldsymbol{c}$ only:

$$\min_{\boldsymbol{c}} \|\mathbf{y} - XD_{\boldsymbol{c}}\hat{\boldsymbol{\gamma}}(\boldsymbol{c})\|^2 + \lambda\|\hat{\boldsymbol{\gamma}}(\boldsymbol{c})\| + \mu \sum_{i=1}^{d} c_i^{2\delta}, \tag{33}$$

and we can optimize it through gradient descent.

To verify that this constraint leads sub-$\ell_1$ pseudo-norm to the original $\boldsymbol{\beta}$, first we reformulate the original objective with the method of Lagrangian multipliers:

$$\mathcal{L}(\boldsymbol{\nu}, \mu) = \|X\boldsymbol{\beta} - \mathbf{y}\|^2 + \sum_{i=1}^{d} \nu_i \beta_i^2 + \mu \sum_{i=1}^{d} \frac{1}{\nu_i^{\delta}} \tag{34}$$

Then, we find stationary points for $\nu_i$ by taking the gradients and setting them to zero:

$$\frac{\partial \mathcal{L}(\boldsymbol{\nu}, \mu)}{\partial \nu_i} = \beta_i^2 - \mu\delta\nu_i^{-(\delta+1)} = 0 \quad \Rightarrow \quad \nu_i^{\delta+1} = \frac{\mu\delta}{\beta_i^2}$$

$$\Rightarrow \quad \nu_i = \left(\frac{\mu\delta}{\beta_i^2}\right)^{\frac{1}{\delta+1}} \tag{35}$$

The penalty in the original adaptive ridge regression can be rewritten as:

$$\sum_{i=1}^{d} \nu_i \beta_i^2 = \sum_{i=1}^{d} \left(\frac{\mu\delta}{\beta_i^2}\right)^{\frac{1}{\delta+1}} \beta_i^2 \propto \sum_{i} \beta_i^{2(1-\frac{1}{\delta+1})}. \tag{36}$$

Note that when $\delta = 1$, we recover the Lasso $\ell_1$ norm. When $\delta \to 0$, it approaches the $\ell_0$ pseudo-norm, which basically counts the number of nonzero entries.

# C NIMO: IMPLEMENTATION DETAILS

In this section, we provide a detailed introduction to the structure of NIMO, along with the strategies we used to improve training.

## C.1 MODEL DEFINITION

We define NIMO as follow:

$$f(\mathbf{x}) = \beta_0 + \sum_{j=1}^{d} x_j \beta_j (1 + g_{\boldsymbol{u}_j}(\mathbf{x}_{-j})) \quad \text{s.t.} \quad g_{\boldsymbol{u}_j}(\mathbf{0}) = 0 . \tag{37}$$

The architecture of our model consists of two components, as illustrated in Fig. 2. The left component is a nonlinear neural network that operates on the input features $\mathbf{x}_{-j}$, where the $j$-th element has been masked out. The right component is a linear model, in which coefficients $\beta_j$ are multiplied by the corresponding neural network outputs and subsequently summed.

In practice, for each feature component $x_j$ of the input vector $\mathbf{x}$, the value is temporarily masked to form $\mathbf{x}_{-j}$, which is then passed through a neural network $g_{\boldsymbol{u}_j}$. The output of this network is added to a constant 1 and multiplied by the original feature value $x_j$, resulting in a rescaled version of the feature:

$$\tilde{x}_j = x_j \cdot (1 + g_{\boldsymbol{u}_j}(\mathbf{x}_{-j})) \tag{38}$$

This rescaling can be interpreted as an element-wise modulation of the input features based on context. The rescaled features are then linearly combined using fixed coefficients $\boldsymbol{\beta}$ to produce the final prediction:

$$y = \beta_0 + \sum_{j=1}^{d} \tilde{x}_j \beta_j.$$

## C.2 PRACTICAL IMPLEMENTATION

We implement our model with PyTorch Lightning (Falcon & The PyTorch Lightning team, 2019). While the model definition is simple, in practice, several strategies are needed to make the training more efficient and effective.

**Positional Encoding** In principle each feature $x_j$ could have its own dedicated network $g_{\boldsymbol{u}_j}$, but this can be computationally expensive for high-dimensional inputs. To reduce overhead, we instead use a shared neural network $g_{\boldsymbol{u}}$ across all components. To ensure the network remains aware of which feature is currently masked out, we append a positional encoding to $\mathbf{x}_{-j}$ before passing it through the shared network. We assign each masked input feature $\mathbf{x}_{-j}$ a unique binary vector corresponding to its index $j$, represented in fixed-length binary form using $\lfloor \log_2 d \rfloor + 1$ bits. This compact encoding allows the shared neural network to remain index-aware with minimal overhead.

**Group Penalty and Noise Injection** In the main paper we showed that linear coefficients $\boldsymbol{\beta}$ in NIMO are interpretable as in a linear model. Even though the neural network contribution is not interpretable per se, we can still identify relevant nonlinear interactions by means of sparsity. The idea is to force the first layer of the neural network to be sparse, hence to use only relevant features. To do so, we apply a group penalty on the weight matrix $W \in \mathbb{R}^{p \times d}$ of the first fully-connected (`fc`) layer, where $d$ is the number of input features and $p$ is the hidden dimension of the first `fc` layer. In order to obtain sparsity at the input feature level, we need to consider neurons connected to the same input feature and penalize them as a group:

$$\ell_{\text{group}} = \lambda_{\text{group}} \sum_{j=1}^{d} \|\mathbf{w}_j\|_2 , \tag{39}$$

where $\mathbf{w}_j = W[:, j] \in \mathbb{R}^p$ is the weight vector acting on the $j$-th feature. With this penalty on the first `fc` layer, we encourage it to select only certain input features. However, since an MLP computes a composition of functions, the subsequent `fc` layers can still compensate for the sparsity induced

by the regularization in the first layer. Specifically, even if the first layer produces sparse outputs, subsequent layers can reweigh and transform these outputs to restore their influence on the final prediction. To mitigate this compensation effect, we inject noise into the output of the first layer. By doing so, we disrupt the deterministic signal path, making it harder for the subsequent layers to simply exploit the resulting weak, regularized, and noisy signal. As a result, the network is encouraged to focus on strong and robust signals associated with informative input features. This approach works very well in practice and allows to easily achieve sparsity in the neural network.

**Other Details** After the second `fc` layer, a `sin` activation function is applied to capture potentially repeating or high-frequency cyclic patterns in the data (Sitzmann et al., 2020). The learning rate is `1e-3` for synthetic experiments, and `5e-3` for real-world datasets. We choose Adam (Kingma, 2014) as the default optimizer. In all experiments, the lasso and group penalties are selected by performing a grid search over a predefined set of candidate values.

# D   EXPERIMENTS

We conduct experiments on both synthetic and real-world datasets. All experiments are conducted on a workstation equipped with a single user-grade NVIDIA GeForce RTX 5080 GPU. The methods we compare with are linear model (Lasso for regression, Logistic Regression for classification), a vanilla neural network (**NN**) (Goodfellow et al., 2016), and several state of the art interpretable approaches, namely **LassoNet** (Lemhadri et al., 2021), Neural Additive Models (**NAMs**) (Agarwal et al., 2021) and Interpretable Mesomorphic Networks (**IMNs**) (Kadra et al., 2024).

## D.1   METHODS IMPLEMENTATION

**Linear Model** We use scikit-learn (Pedregosa et al., 2011) to implement the Lasso and Logistic regression model. More specifically, we use `LassoCV` and `LogisticRegressionCV` with default 5-fold cross validation to select the best model. For `LogisticRegressionCV`, the $\ell_1$ penalty is applied.

**Neural Network** We implement the naive neural network (NN) with PyTorch Lightning (Falcon & The PyTorch Lightning team, 2019). For a fair comparison, the NN has the same structure as the nonlinear part in our NIMO model, except no positional encoding. We use Adam (Kingma, 2014) as the default optimizer. The learning rate is `1e-3` for synthetic experiments, and `5e-3` for real-world datasets. Strong dropout regularization (`p=0.6`) is used to avoid overfitting for small scale datasets (including synthetic datasets, diabetes and Bostong housing datasets). For the relatively large superconductivity dataset, small dropout rate (`p=0.1`) is used.

**LassoNet** We use the original implementation of LassoNet from their GitHub repository [2]. Using the same API as shown in the LassoNet document, we configure the model to have a structure and number of parameters comparable to our model. All other hyperparameters are kept at their default values. The best model is automatically selected along the entire regularization path, as demonstrated in their paper. The one with the smallest validation loss is chosen.

**NAM** The official NAM implementation was in Tensorflow. We chose a well-maintained PyTorch implementation from the GitHub repository [3]. This implementation provides a scikit-learn style interface, which can be easily used.

**IMN** We use the original PyTorch implementation of IMN from their GitHub repository [4]. All the arguments are kept at their default values except modifying the data loading logic to fit our datasets.

For all the methods above, we tried our best to make sure that they achieve reasonably good and consistent performance on all the datasets.

---

[2]https://github.com/lasso-net/lassonet
[3]https://github.com/lemeln/nam
[4]https://github.com/ArlindKadra/IMN

## D.2 SYNTHETIC EXPERIMENTS: SETUP

We create the synthetic datasets by first sampling the features, and then compute the targets according to different nonlinear patterns. Random Gaussian noises are added to the targets. The feature values are sampled uniformly between $-2$ and $2$. For logistic regression, we pass the targets through a logistic (sigmoid) function to get probabilities between $0$ and $1$, and then sample the final binary labels using a Bernoulli distribution with those probabilities. In all the settings, we use 200 training samples and 100 samples each for validation and test.

**Settings for regression** All features are indexed starting from 1. $\epsilon \sim \mathcal{N}(0, 0.1^2)$ is a Gaussian noise.

- Setting 0 (toy example), $n = 200, p = 3$

$$
\begin{aligned}
y = \ & + 3 \cdot x_1 \cdot [1 + \tanh(10x_2)] \\
& - 3 \cdot x_2 \cdot [1 + \sin(-2x_1)] \\
& + \epsilon
\end{aligned}
\tag{40}
$$

- Setting 1, $n = 200, p = 5$

$$
\begin{aligned}
y = \ & + 3 \cdot x_1 \cdot [1 + (2\sigma(x_2 x_3) - 1)] \\
& - 2 \cdot x_2 \\
& + 2 \cdot x_3 \\
& + \epsilon
\end{aligned}
\tag{41}
$$

- Setting 2, $n = 200, p = 10$

$$
\begin{aligned}
y = \ & + 1 \cdot x_1 \cdot [1 + \tanh(x_2 x_3 + \sin(x_4))] \\
& + 2 \cdot x_2 \cdot [1 + \sin(2x_1)] \\
& - 1 \cdot x_3 \cdot \left[1 + \frac{2}{\pi} \arctan(x_2 x_4)\right] \\
& + \epsilon
\end{aligned}
\tag{42}
$$

- Setting 3, $n = 200, p = 50$

$$
\begin{aligned}
y = \ & - 2 \cdot x_1 \cdot [1 + \tanh(x_2 x_4)] \\
& + 2 \cdot x_2 \cdot \left[1 + \frac{2}{\pi} \arctan(x_4 - x_5)\right] \\
& + 3 \cdot x_4 \cdot [1 + \tanh(x_2 + \sin(x_5))] \\
& - 1 \cdot x_5 \cdot [1 + (2\sigma(x_1 x_4) - 1)] \\
& + \epsilon
\end{aligned}
\tag{43}
$$

- Setting 4 (vanilla regression), $n = 200, p = 10$

$$
\begin{aligned}
y = \ & -5 \cdot x_1 - 4 \cdot x_2 - 3 \cdot x_3 - \cdots + 0 \cdot x_6 \\
& + 1 \cdot x_7 + 2 \cdot x_8 + \cdots + 4 \cdot x_{10} \\
& + \epsilon
\end{aligned}
\tag{44}
$$

**Settings for classification** In order to create data for synthetic classification we follow a similar procedure as in the regression cases, where we directly control the linear coefficients and the nonlinear interactions. The only difference is that we further apply a link function and binarize the output and get the labels. Note that this way we loose information about the original linear coefficients, which cannot be recovered exactly anymore. The sparsity in the coefficients still remains crucial. For all the settings, we generate $y$ first, and then generate labels through the following procedure:

$$
\pi = \frac{1}{1 + e^{-y}}, \quad \text{label} \sim \text{Bernoulli}(\pi)
$$

Table 3: Classification accuracy for synthetic classification settings. We compare with logistic regression, NN and LassoNet. In each setting we use 200 samples.

|  | Features | Log. Regr. | NN | NIMO | LassoNet |
|---|---|---|---|---|---|
| Setting 1 | 3 | 0.59 | 0.90 | **0.92** | 0.89 |
| Setting 2 | 10 | 0.68 | 0.83 | **0.85** | 0.82 |
| Setting 3 | 50 | 0.74 | 0.69 | 0.84 | **0.90** |

- Setting 1, $n = 200, p = 3$

$$
\begin{aligned}
y = \ & + 2 \cdot x_1 \cdot [1 + 2 \tanh(x_2)] \\
& - 2 \cdot x_2 \cdot [1 + 3 \sin(2x_1) + \tanh(2x_1)] \\
& + 1 + \epsilon
\end{aligned}
\tag{45}
$$

- Setting 2, $n = 200, p = 10$

$$
\begin{aligned}
y = \ & + 10 \cdot x_1 \cdot [1 + 2 \cdot \tanh(2x_2) + \sin(x_4)] \\
& + 20 \cdot x_2 \cdot [1 + 2 \cdot \cos(2x_1)] \\
& - 20 \cdot x_3 \cdot [1 + 2 \cdot \arctan(x_2 x_4)] \\
& + 10 \cdot x_4 \\
& - 10 + \epsilon
\end{aligned}
\tag{46}
$$

- Setting 3, $n = 200, p = 50$

$$
\begin{aligned}
y = \ & - 20 \cdot x_1 \cdot [1 + \tanh(x_2 x_4)] \\
& + 20 \cdot x_2 \cdot \left[1 + \frac{2}{\pi} \arctan(x_4 - x_5)\right] \\
& + 30 \cdot x_4 \cdot [1 + \tanh(x_2 + \sin(x_5))] \\
& - 10 \cdot x_5 \cdot [1 + (2\sigma(x_1 x_4) - 1)] \\
& + \epsilon
\end{aligned}
\tag{47}
$$

### D.3 SYNTHETIC EXPERIMENTS: ADDITIONAL RESULTS

Below, we present additional experimental results for the synthetic settings described above.

**Regression** As we mentioned in the main paper, we further apply group $\ell_2$ regularization to the weight matrix of the first fully connected (fc) layer, encouraging feature-level sparsity and clarifying how inputs contribute to predictions. We provide the details in Appendix C.2. Here, we show the sparse features selected by the neural network in Figure 8. We observe that even though all features are input to the neural network, only a small subset of features are selected. Moreover, these selected features align with those involved in the nonlinearity computation in our settings, which demonstrates the model's ability to effectively identify and focus on the most relevant features for the task.

**Classification** Our method can be extended to generalized linear models and here we showcase it for logistic regression. Also in this case we show various settings for different dimensionality, nonlinear terms and different number of uninformative features. In Table 3 we report the classification accuracy on the test set for all methods. Overall, our model significantly outperforms logistic regression on all settings. When analyzing the learned coefficient, we can see that only NIMO recovers the correct sparsity patterns. As argued before, this allows to learn the correct nonlinear interactions. Other nonlinear models and hybrid approaches perform similarly to NIMO but fail to provide the interpretability of the coefficients as in a linear model.

In classification, the presence of the link function can obscure the underlying signal, so while a linear decision boundary may suffice, recovering the true generative coefficients is more difficult than in regression, where the model directly learns to approximate the target surface. Across all settings, the goal is not to recover the precise magnitudes of the ground truth coefficients, but to accurately identify their sparsity pattern and directional effects (i.e., signs). We compare the recovered coefficients from

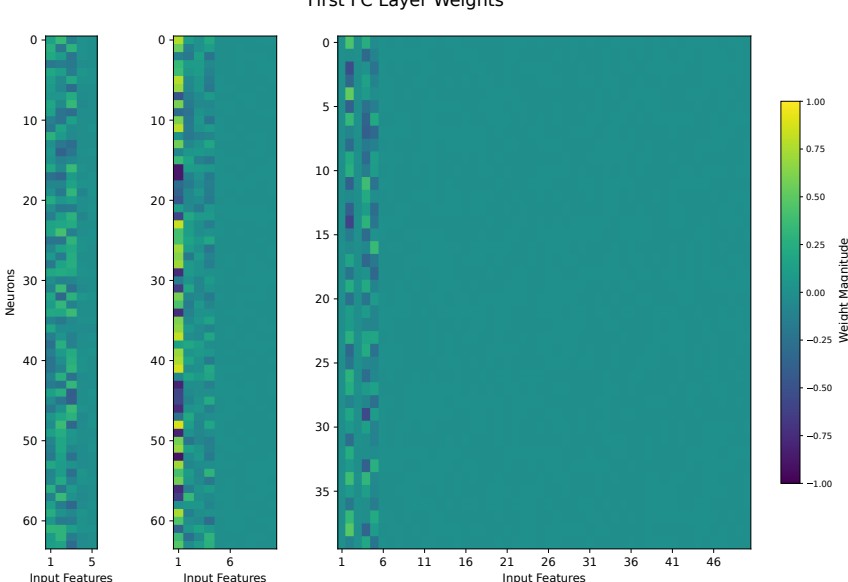

Figure 8: Sparsity and feature selection in the first `fc` layer of NIMO for regression settings. *Left*: Setting 1. *Middle*: Setting 2. *Right*: Setting 3.

logistic regression and our model across three classification settings, as shown in Figure 9. Since recovering the exact magnitudes of the coefficients is not feasible, we normalize all coefficients, including the ground truth, by dividing them by the maximum absolute value among the coefficients, to enable a fair comparison.

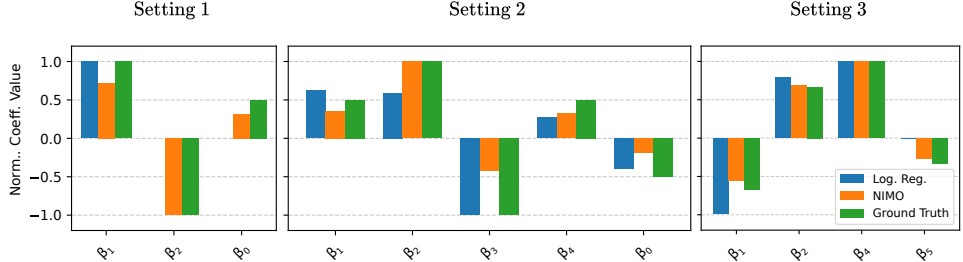

Figure 9: Learned coefficients for synthetic classification settings. We compare NIMO and Logistic regression with the ground truth. Since we use binarized labels it is impossible to retrieve the exact coefficients. Instead, we compare normalized coefficients, such that the maximum is always 1.

We observe that logistic regression struggles to identify the relevant features and their correct signs. In contrast, our model generally succeeds in selecting the correct features, and the neural network component effectively captures sparse but important features that contribute to nonlinearity. It is worth noting that NIMO also selects some noisy features, especially in Setting 3. However, our experiments are primarily intended as a proof of concept to demonstrate that our model can be easily extended to generalized linear models. With careful hyperparameter tuning, NIMO is capable of selecting more accurate features. The sparse features selected by the neural network are shown in Figure 10.

### D.4 REAL-DATASET EXPERIMENTS

We conduct experiments on three real-world datasets: the diabetes dataset (Efron et al., 2004), the Boston housing dataset (Belsley et al., 2005) and the superconductivity dataset (Hamidieh, 2018). Each dataset is randomly split into training (60%), validation (20%), and test (20%) sets, repeated

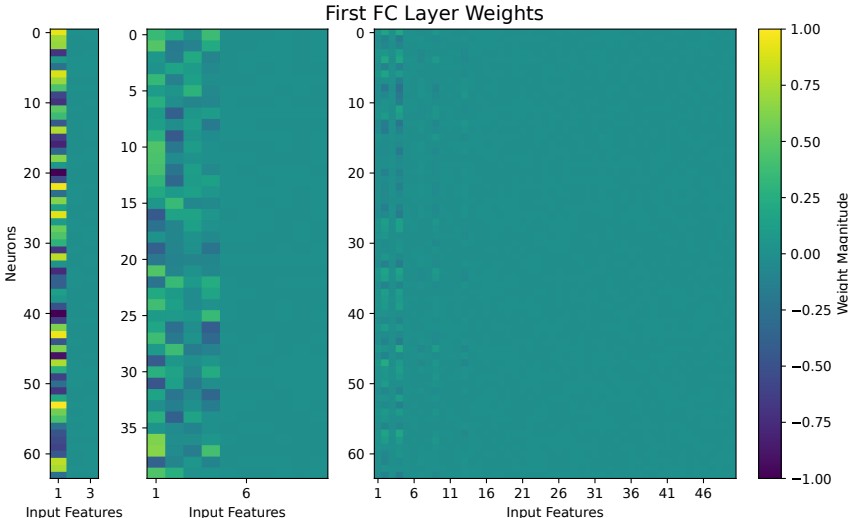

Figure 10: Sparsity and feature selection in the first `fc` layer of NIMO for classification settings. *Left*: Setting 1. *Middle*: Setting 2. *Right*: Setting 3.

over 5 independent runs. For each split, we perform a grid search to select the optimal penalty parameter based on the validation loss (or validation accuracy for classification tasks). Error bars and confidence intervals are reported across these different splits. The main results are shown in Figure 5.

**Diabetes dataset** The diabetes dataset is a classic regression dataset consisting of 442 patients, each with 10 numeric features, and the target measures disease progression one year after baseline. As shown in Figure 11 (left), we observe that all methods achieve similar MSE losses and the neural network (also the hybird methods) shows no clear advantage on this dataset. Since the diabetes dataset is commonly used for linear regression tasks, it is not surprising that the data does not exhibit complex nonlinear interactions among features. We can verify this claim by looking at the weight sparsity of the first `fc` layer in NIMO, shown in Figure 11 (right). Relevantly, NIMO achieves a significantly sparser solution than Lasso regression, see Figure 11 (middle). This is particularly evident by looking at the coefficients associated with the feature "s1", "s2" and "s4".

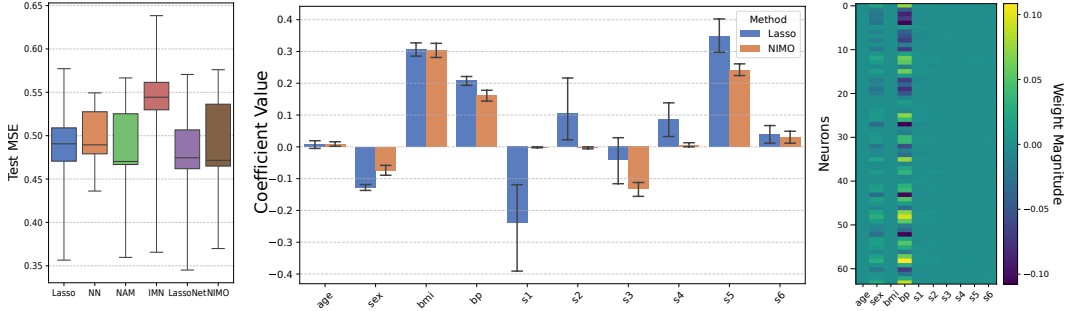

Figure 11: Diabetes datasets results. *Left*: MSE loss on test set. All methods achieve comparable results. *Middle*: Learned linear coefficients by Lasso and NIMO. NIMO selects sparser coefficients ("s1, "s2", "s4"). *Right*: the first `fc` layer of NIMO is very sparse; few nonlinearities are relevant.

**Boston dataset** The Boston housing dataset contains 506 instances, each with 13 features, and the target is the median value of owner-occupied homes. As shown in Figure 12 (left), our method performs significantly better than Lasso regression and shows compatible MSE test loss with the other nonlinear and hybrid approaches. This suggests that nonlinear interactions are crucial to explain the data. While the other methods are able to capture those nonlinearities, they lack the interpretability provided by our model. It is important to note that the Boston housing dataset has been criticized

for containing controversial features [5]. The feature of interest is termed "B". From the coefficient plot in Figure 12 (middle), we observe that the significance of the feature "B" is greatly reduced in our model compared to Lasso regression. This surprising result suggests that the role of the feature "B" towards the prediction might be irrelevant. Note that the feature "B" is also not relevant for the nonlinear terms, as shown in Figure 12 (right).

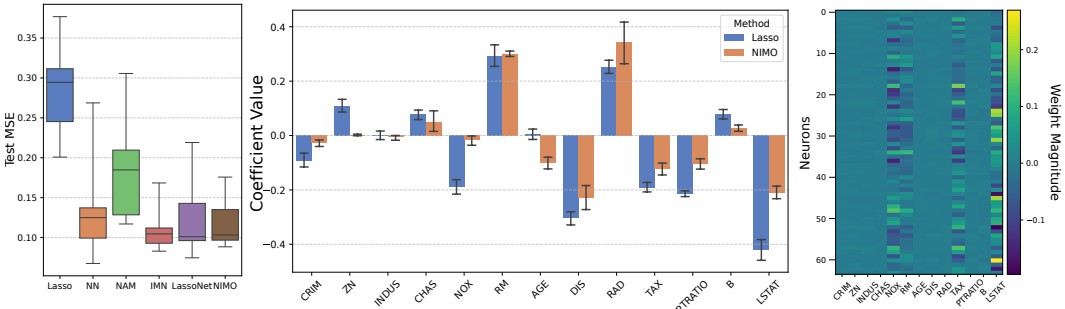

Figure 12: Boston datasets results. *Left*: Comparison of MSE losses. NIMO performs on par with other methods and significantly outperforms Lasso, suggesting nonlinear interactions are crucial. *Middle*: Learned linear coefficients by Lasso and NIMO. The two models learn coefficient with a comparable sparsity. Relevantly, the feature "B" is almost irrelevant for NIMO. *Right*: sparsity in the first `fc` layer of NIMO.

**Superconductivity** The superconductivity dataset contains 21,263 instances, each with 81 features. The target is to predict the critical temperature based on the features extracted. In Figure 13, we

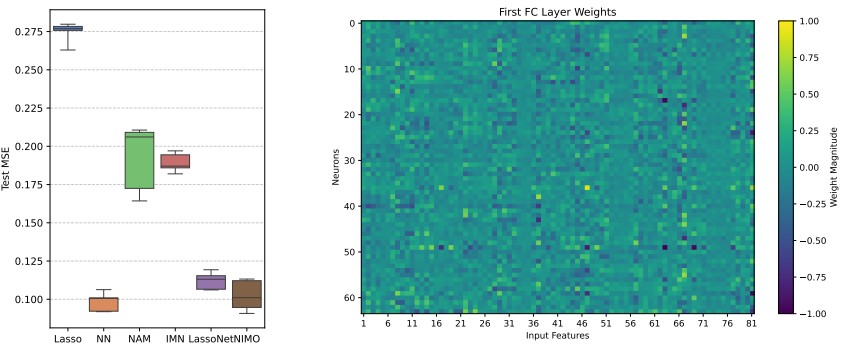

Figure 13: Superconductivity dataset results. *Left*: Comparison of MSE losses. *Right*: sparsity in the first `fc` layer of NIMO.

observe that NIMO achieves the best performance comparable to general neural network. With respect to sparsity in the first `fc` layer, the presence of strong correlations among several of the 81 features makes it challenging to obtain clear sparsity patterns. Nevertheless, certain patterns can still be discerned (Figure 13, right). Figure 14 shows the coefficients for Lasso and NIMO on this dataset.

### D.5 ABLATION STUDY OF FEATURE MASKING

To verify the effectiveness of masking out the $j$-th component of an instance when passing it through the neural network, we conducted an ablation study. Specifically, we examined the effect of masking out the $j$-th component as well as the effect of removing position encoding. From the ablation results across the three settings (Figure 7, Figure 15, and Figure 16), we observe several consistent patterns:

---

[5]https://fairlearn.org/main/user_guide/datasets/boston_housing_data.html

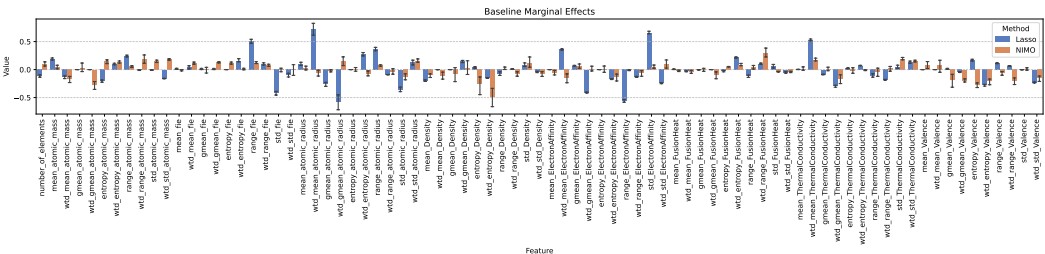

Figure 14: Coefficients (also MEMs) for Lasso and NIMO on superconductivity dataset.

- Without masking the $j$-th component, the first `fc` layer of the neural network can still exhibit sparsity patterns, but they are less distinct than when masking is applied. We attribute this to the position encoding, which enables the network to identify the feature index.

- Without masking the $j$-th component, the magnitudes of the linear coefficients are distorted, reducing the interpretability of the marginal effects at the mean (MEM).

- Further removing the position encoding distorts the sparsity patterns in the first `fc` layer and substantially alters the magnitude of the linear coefficients, thereby undermining interpretability.

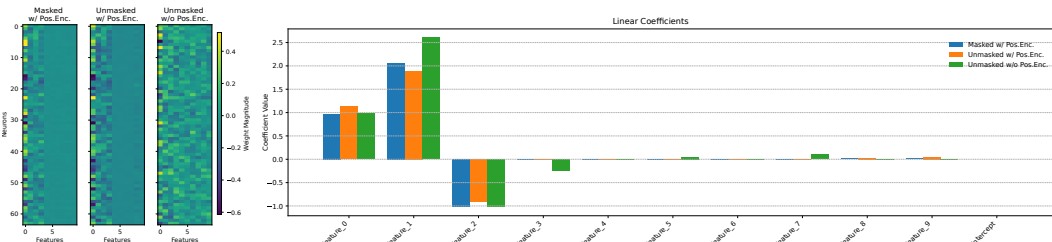

Figure 15: The effect of feature masking and position encoding on the linear coefficients and the sparsity of the first `fc` layer on setting 2.

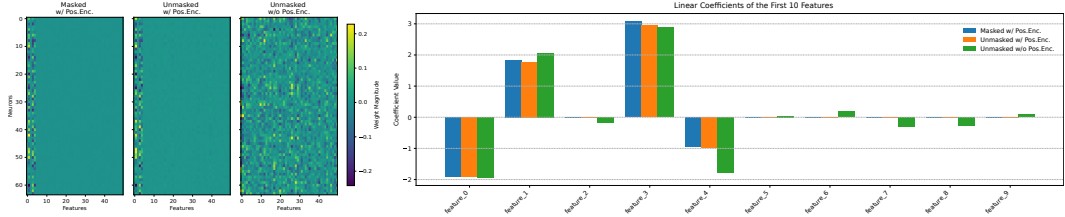

Figure 16: The effect of feature masking and position encoding on the linear coefficients and the sparsity of the first `fc` layer on setting 3. Coefficients are displayed for the first 10 features.

## D.6 LIMITATIONS OF NIMO

To clearly and explicitly illustrate the limitations of our model, we conduct an additional set of ablation studies in this subsection. As discussed above, due to its architectural constraints, NIMO is not able to capture datasets with strong nonlinearity or purely self-interacting features. To demonstrate this, we create the following synthetic dataset:

$$y = x_1^2 + x_2^2 + x_3^2 + \epsilon, \tag{48}$$

where the feature values are sampled uniformly from $[-2, 2]$ and $\epsilon \sim \mathcal{N}(0, 1)$ is a Gaussian noise. We compare the predictive performance of Lasso, NIMO, and LassoNet in Table 4. As expected, NIMO performs similarly poor to Lasso, while LassoNet is still able to model this dataset effectively. From

Figure 17 (*left*), we observe that the coefficients learned by NIMO closely resemble those of Lasso, and NIMO effectively learns only a constant intercept $\beta_0$. In addition, we plot the weight matrix of the first `fc` layer of the neural network and observe that all weights are close to zero. This further confirms that, in this setting, NIMO reduces to a purely linear model without contributing additional nonlinear modeling capacity. This outcome is fully consistent with our expectations and illustrates a known limitation of NIMO: it cannot recover self-nonlinearities such as $x_j^2$ due to its interpretability-preserving architectural design. Although LassoNet is able to model this type of nonlinearity, it does not offer the same ability as NIMO or linear models to provide global, population-level interpretability through stable and directly interpretable learned coefficients.

Table 4: Predictive performance on the synthetic self-interaction dataset. NIMO performs similarly to Lasso in the original feature space but succeeds after basis expansion, where quadratic terms are explicitly included.

| Method (#features) | Original (3) | Basis Expansion (9) |
|---|---|---|
| Lasso | 5.307 | 1.009 |
| LassoNet | **1.027** | 1.012 |
| NIMO | 5.340 | **1.009** |

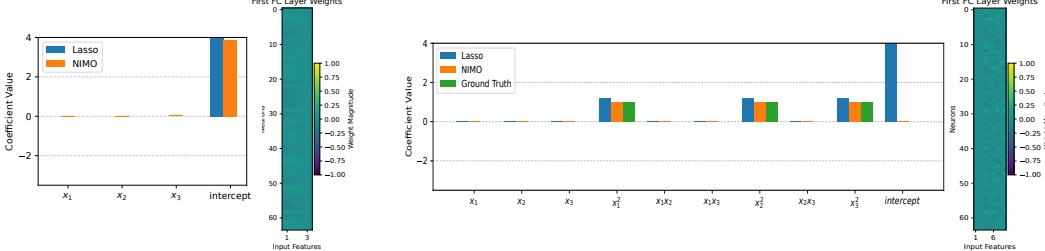

Figure 17: Learned coefficients and first-layer weights on the self-interaction dataset. (*Left*) In the original feature space, NIMO collapses to a constant model, matching Lasso. (*Right*) With basis expansion, NIMO recovers the quadratic effects.

However, if we know or suspect that certain feature interactions play a role in the dataset, we can model them explicitly through feature engineering. In this experiment, we apply a second-order polynomial basis expansion, transforming the original feature space $[x_1, x_2, x_3]$ into $[x_1, x_2, x_3, x_1^2, x_1 x_2, x_1 x_3, x_2^2, x_2 x_3, x_3^2]$. From Table 4, we can see that after basis expansion, both NIMO and Lasso achieve performance comparable to LassoNet. In Figure 17 (*right*), NIMO also correctly recovers the linear coefficients associated with the quadratic terms $x_1^2$, $x_2^2$, and $x_3^2$, demonstrating that NIMO can model self-nonlinearities once they are explicitly included in the feature representation. In summary, while NIMO cannot capture self-interactions in the raw feature space due to its interpretability-preserving architecture, simple basis expansions allow it to model such nonlinearities effectively while retaining global interpretability.

### D.7 LOCAL EXPLANATIONS

NIMO provides intelligible global interpretability through the marginal effects at the mean (MEM) and per-instance adjustments via the correction networks. In the sections above, we have demonstrated the effectiveness of NIMO's global interpretation. Here, we briefly showcase its ability to provide meaningful local, per-instance explanations.

**Local Explanation on Synthetic Dataset.** For local explanations, NIMO defines a *local coefficient* for each feature,

$$s_j(\mathbf{x}) = \beta_j\big(1 + g_{\mathbf{u}_j}(\mathbf{x}_{-j})\big), \tag{49}$$

which represents how the global effect $\beta_j$ is modulated by the specific input context. Based on this, we compute the *local contribution*

$$c_j(\mathbf{x}) = x_j\, s_j(\mathbf{x}) \tag{50}$$

for each feature at each instance. The scatter plots in Figure 18 compare the ground-truth contribution term for feature $j$ with NIMO's estimated local contribution $c_j(\mathbf{x})$ for synthetic setting 1. In this setting, the dataset contains five features, but only the first three have nonzero linear coefficients in the data-generating process. As shown in the figure, the first three features exhibit scatter points that align closely with the identity line, while the remaining two features produce scatter distributions centered near the origin. These results indicate that NIMO's local explanations faithfully capture the underlying structure of the data.

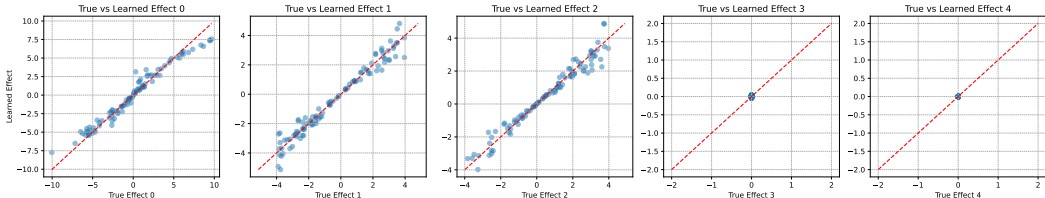

Figure 18: Local explanations for each feature on the synthetic setting 1.

**Local Explanation on Real Dataset.** For real datasets, where ground-truth local contributions are unavailable, we compare NIMO's local contributions $c_j(\mathbf{x})$ with SHAP values. As shown in Figure 19, the scatter plots for most features align closely with the identity line, indicating that NIMO's local explanations are consistent with widely used post-hoc explanation tools while remaining **intrinsic** to the model. Due to the sparsity constraint, some coefficients learned by NIMO are exactly zero, which forces the corresponding local contributions to be zero as well. In such cases, the scatter plot reflects this behavior clearly. For example, the local contributions for the feature NOX collapse to zero for all instances.

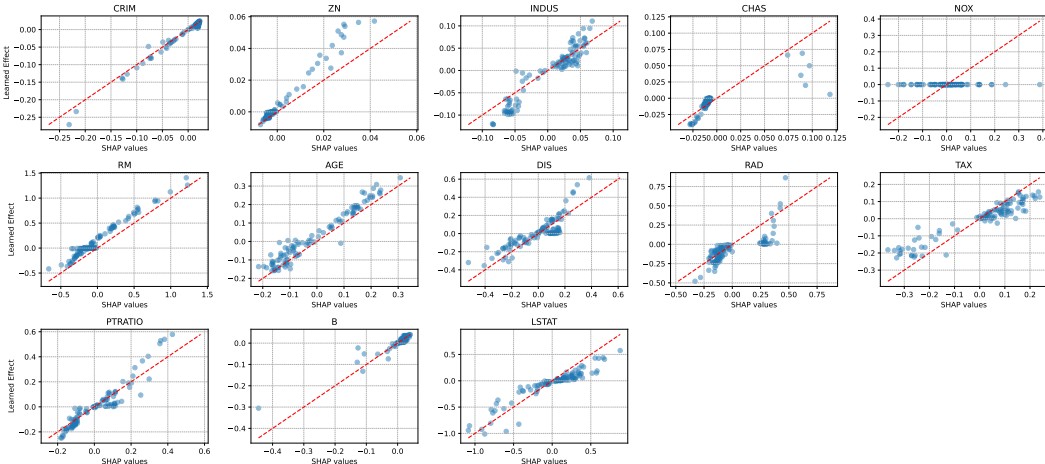

Figure 19: Local explanations for each feature on the Boston housing dataset.

### D.8 COMPARISON OF THE DEGREE OF NONLINEARITY

In the main paper (Sec. 4, Figure 5), we observed a substantial difference in predictive performance between the Diabetes and Boston Housing datasets, and attributed this phenomena to the relatively low level of nonlinearity present in the Diabetes dataset. To experimentally and quantitatively verify this claim, we compute the absolute magnitude of the nonlinear correction for each feature,

$$n_j(\mathbf{x}) = \left| 1 + g_{\mathbf{u}_j}(\mathbf{x}_{-j}) \right|, \tag{51}$$

and then average these values over all test samples for each feature. Figure 20 shows the resulting mean nonlinearity levels for both datasets. The Boston Housing dataset exhibits noticeably larger deviations from 1 across many features, indicating stronger nonlinear interactions. In contrast, the

Diabetes dataset shows values much closer to 1 for all features, confirming that its underlying structure is primarily linear. These results support our interpretation of the performance differences observed in the main experiments.

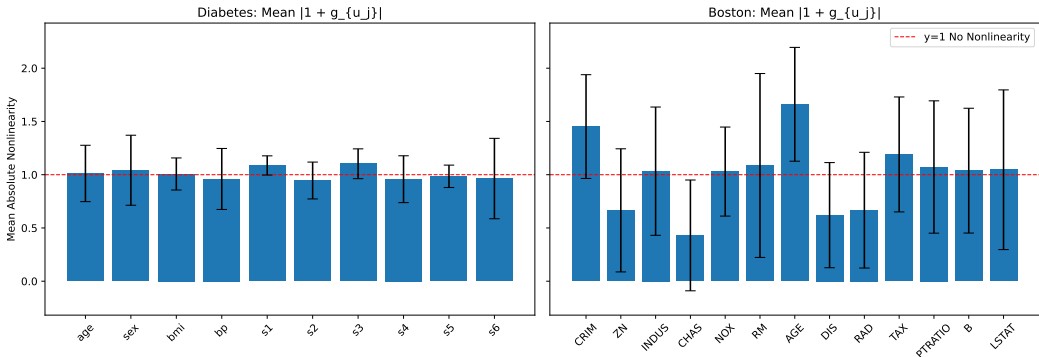

Figure 20: Comparison of the degree of nonlinearity in the Diabetes and Boston Housing datasets.

### D.9 ABLATION STUDY OF THE SPARSE PENALTY ON THE FIRST FC LAYER

To verify whether applying the group penalty to the first `fc` layer of the neural network is beneficial, we conduct an ablation study on the synthetic dataset (setting 1). From Figure 21, we can see that

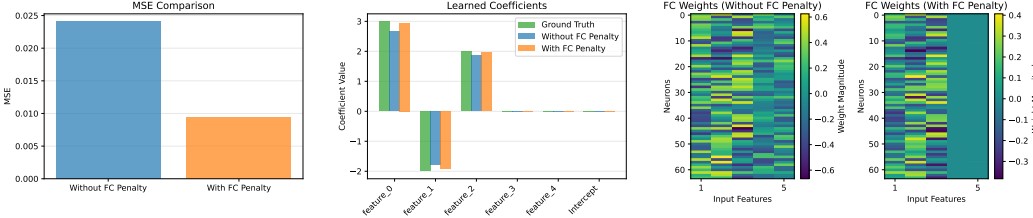

Figure 21: Ablation study of the group penalty on the first `fc` layer.

removing the group penalty on the first `fc` layer leads to worse final performance. Furthermore, comparing the learned coefficients shows that they deviate more from the ground truth. Inspecting the weight matrix of the first `fc` layer, we also observe that the neural network begins to use the two redundant features, `feature3` and `feature4`, even though they were not used in the data generation process for setting 1 (see D.2).

### D.10 EXTRA RESULTS ON PMLB BENCHMARK

We conduct a more extensive evaluation of prediction performance using the PMLB benchmark Romano et al. (2021). We filter out datasets that were either too small (less than 1,000) or too large (more than 16,000), resulting in a total of 10 regression datasets. For each dataset, we compare the performance of Lasso, LassoNet, a standard Neural Network, and NIMO. Hyperparameters for all models are selected via grid search. The results are presented in Figure 22.

As discussed in the limitations of NIMO (Sec. D.6), some datasets require additional feature engineering for NIMO to perform well. Within the PMLB benchmark, we indeed identified such a case. The `225_puma8NH` dataset cannot be well fitted by NIMO directly (yielding an MSE of around 0.4). However, after applying a polynomial basis expansion, NIMO achieves performance comparable to LassoNet and Neural Networks.

## E   LLM STATEMENT

ChatGPT is used only for polishing the sentences.

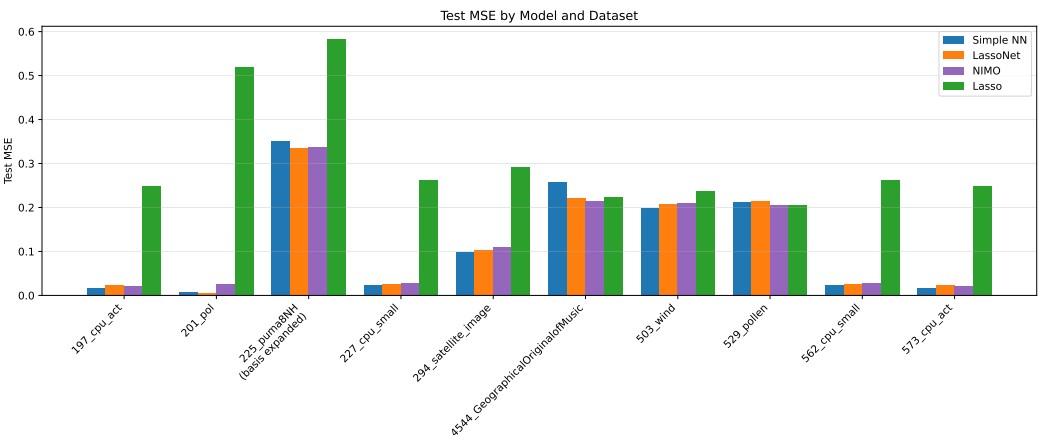

Figure 22: Comparison of prediction performance on PMLB.

