# OpenReview forum: "NIMO: a Nonlinear Interpretable MOdel"
_ICLR.cc/2026/Conference — ICLR 2026 Poster_

### Official Review · Reviewer_DuVs · 2025-10-15

**Soundness:** 3
**Presentation:** 3
**Contribution:** 2
**Rating:** 4
**Confidence:** 3

**Summary:**

The authors introduce NIMO (Nonlinear Interpretable MOdel), a model that fits a neural network to learn non-linear corrections to a linear model's predictions. The model optimizes the linear coefficients analytically while optimize over the neural network parameters. At inference time, the model can be interpreted using the linear parameters. In a toy example, synthetic experiments, and two tabular datasets, NIMO outperforms some standard baselines (Lasso, LassoNet, and a neural net); in real-world datasets it performs on par with these baselines.

**Strengths:**

- S1: The authors tackle an important problem
- S2: The authors approach is novel
- S3: The authors' optimization approach seems generally reasonable

**Weaknesses:**

- W1: The main weakness seems to be that the model does not provide significant global interpretability, as the model coefficients are a nonlinear function of the input. Thus, the model's provides interpretability similar to instance-level posthoc methods such as LIME or SHAP. The authors should provide a deeper discussion of this and a quantitative comparison showing a scenario where NIMO can provide more interpretability than post-hoc baselines (rather than just a qualitative comparison).
- W2: The real-dataset experiments are fairly limited, it would be nice to see a more substantial evaluation of prediction performance on popular tabular datasets, e.g. the Tab-Arena benchmark or PMLB.
- W3: The authors discuss how NIMO reaches a sparser solution than a Lasso baseline (e.g. Fig 4), but it is unclear whether this is useful as NIMO can simply use the non-linear NN to use these features

**Questions:**

The rationale for excluding a feature's values when computing its coefficient seems unclear. A nonlinear network should be able to effectively deduce the feature's value anyhow, especially if there are any correlations between features.

Similarly, the sparsity/noise added to the first layer of the FC network seems strange, as the network can still recombine different features in any nonlinear manner. Can the authors perform some ablations showing the effect of this?

---

> ### Author Response · Authors · 2025-11-19
>
> We thank the reviewer for their valuable feedback, for highlighting potential misunderstanding in the design of the model and concerning global vs local interpretability. As part of the rebuttal we tried to address all the raised concerns either in text or with new evaluations.
>
> ## Weaknesses:
> > W1: The main weakness seems to be that the model does not provide significant global interpretability, as the model coefficients are a nonlinear function of the input. Thus, the model's provides interpretability similar to instance-level posthoc methods such as LIME or SHAP. The authors should provide a deeper discussion of this and a quantitative comparison showing a scenario where NIMO can provide more interpretability than post-hoc baselines (rather than just a qualitative comparison).
>
> We believe there is a misunderstanding, and we thank the reviewer for the opportunity to clarify.
> 1. In NIMO, the coefficients $\beta$ are **global parameters**, completely independent of the input $x$. Only the multiplicative correction terms $1 + g_{u_j}(x_{-j})$ vary across instances. Consequently, the marginal effect at the mean (MEM) for each feature is **exactly** $\beta_j$, yielding a single, stable global quantity directly analogous to the coefficients in linear regression. This is fundamentally different from instance-level post-hoc methods (e.g., SHAP, LIME) or architectures such as IMN or contextual Lasso, where the effective coefficients change with $x$ and thus do not offer sample-independent global interpretability.
> 2. Most of our experimental analysis is devoted to illustrating NIMO’s **global** interpretability through these $\beta$ coefficients. In the revised version, we also include a dedicated section (Appendix D.7) discussing NIMO’s **local, per-instance** explanations, complementing the global perspective and clarifying the distinction from purely post-hoc approaches.
>
>
> > W2: The real-dataset experiments are fairly limited, it would be nice to see a more substantial evaluation of prediction performance on popular tabular datasets, e.g. the Tab-Arena benchmark or PMLB.
>
> We thank the reviewer for this suggestion. We are still working on the experiments and will include the completed results in the revised version of the paper.
>
> > W3: The authors discuss how NIMO reaches a sparser solution than a Lasso baseline (e.g. Fig 4), but it is unclear whether this is useful as NIMO can simply use the non-linear NN to use these features
>
> This point is closely related to the concern raised by Reviewer F6aQ, and we appreciate the opportunity to clarify.
> 1. We agree that a feature with $\beta_j = 0$ may still contribute to the prediction through the nonlinear networks. This behavior is intentional: allowing $x_j$ to influence the correction terms increases the expressive capacity of the model and enables NIMO to represent cross-feature nonlinear interactions.
> 2. However, this does not diminish the usefulness of sparsity in $\beta$. The coefficients $\beta_j$ coincide exactly with the marginal effects at the mean (MEM) and therefore capture **global, population-level feature influence**. Sparsity in $\beta$ yields a concise and stable global interpretation analogous to that of a sparse linear model.
> 3. While $\beta$ alone is not a complete mechanism for feature selection in NIMO, combining global sparsity in $\beta$ with the magnitude of the coefficient perturbations $\lvert 1 + g_{u_j}(x_{-j}) \rvert$ provides a more holistic view. A feature with $\beta_j = 0$ and perturbation values consistently close to 1 contributes negligibly both globally and locally, and can be safely removed.

---

> > ### Author Response · Authors · 2025-11-19
> >
> > ## Questions:
> > > The rationale for excluding a feature's values when computing its coefficient seems unclear. A nonlinear network should be able to effectively deduce the feature's value anyhow, especially if there are any correlations between features.
> >
> > We appreciate the reviewer’s question and clarify below why masking is a necessary structural component of NIMO’s interpretability guarantees:
> > 1. Without masking, $\beta_j$ loses its interpretation as the marginal effect at the mean (MEM). NIMO’s theoretical guarantee relies on the identity $MEM_j = \beta_j$, which holds only because the nonlinear correction network $g_{u_j}(x_{-j})$ is not allowed to use $x_j$.
> > 2. If $x_j$ were fed into the correction network, the nonlinear term could arbitrarily transform or re-scale $x_j$, leading to $\frac{\partial f}{\partial x_j}\big|_{x=\bar{x}} \neq \beta_j$, and breaking the interpretability of $\beta_j$ as a global additive effect. Masking is therefore a necessary structural constraint to ensure identifiability of $\beta_j$ and to maintain NIMO’s global interpretability guarantee. We also have an ablation study in Sec 4.3 to verify this.
> > 3. Regarding feature correlations: this issue is not unique to NIMO. Lasso and many other interpretable models also experience instability when predictors are highly correlated. In practice, common preprocessing strategies, such as correlation analysis, feature grouping, or dimensionality reduction, can help mitigate this challenge.
> >
> > > Similarly, the sparsity/noise added to the first layer of the FC network seems strange, as the network can still recombine different features in any nonlinear manner. Can the authors perform some ablations showing the effect of this?
> >
> > We thank the reviewer for highlighting this.
> > 1. As discussed in Appendix C.2, we can still promote the identification of relevant nonlinear interactions through sparsity (although the nonlinear components of the network are not directly interpretable). By encouraging sparsity in the first layer, we bias the network to rely primarily on informative input features when modeling interactions.
> > 2. However, sparsifying only the first layer is not always sufficient: deeper layers may compensate by reweighting or reconstructing information from weak signals that pass through the sparse layer. To mitigate this compensation effect, we introduce noise injection after the first layer. This disrupts the deterministic signal path and makes it more difficult for subsequent layers to exploit faint or noisy activations. As a result, the network is encouraged to rely on strong, stable signals corresponding to genuinely relevant features.
> > 3. In the revised version, we include an ablation study examining the effect of the group-sparse penalty on the FC layer (Appendix D.9). Regarding noise injection, our empirical results show that it is beneficial primarily when the group-sparse penalty is relatively weak. Once the group-sparse penalty is sufficiently strong and effectively drives irrelevant features to zero, the additional effect of noise injection becomes negligible. In the training scripts, we also provide the choice to enable or disable noise injection.

---

> > > ### Author Response · Authors · 2025-11-21
> > >
> > > > W2: The real-dataset experiments are fairly limited, it would be nice to see a more substantial evaluation of prediction performance on popular tabular datasets, e.g. the Tab-Arena benchmark or PMLB.
> > >
> > > We thank the reviewer for this suggestion. We have expanded our empirical evaluation to include additional real-world datasets from the PMLB benchmark. The revised version (Appendix D.10, Figure 22) now reports these results. Across the benchmark, NIMO achieves predictive performance comparable to LassoNet and standard neural networks, demonstrating the competitiveness of our approach on diverse tabular datasets.

---

> > > > ### Comment · Reviewer_DuVs · 2025-11-25
> > > >
> > > > I thank the reviewers for their comments, but maintain my score and concerns.

---

> > > > > ### Author Response · Authors · 2025-11-27
> > > > >
> > > > > We thank the reviewer again for the time spent evaluating our work and for the follow-up response. In our rebuttal and the revised manuscript, we have tried to exhaustively address the concerns raised and further added the requested benchmark experiments (Appendix D.10, Figure 22).
> > > > >
> > > > > We would appreciate a clarification on which specific concerns remain unresolved and whether there are particular points that still require attention. That would be very helpful for further improving our work.

---

### Official Review · Reviewer_F6aQ · 2025-10-29

**Soundness:** 3
**Presentation:** 3
**Contribution:** 2
**Rating:** 4
**Confidence:** 3

**Summary:**

A new interpretable hybrid neural network architecture is presented. The architecture differs from existing ones by perturbing the weights of a linear model via a neural network. A novel optimization algorithm is presented to fit said architecture, and it is compared to state-of-the-art hybrid neural networks on synthetic and real tabular datasets.

**Strengths:**

The optimization algorithm consists of expressing the linear coefficients as the closed-form solution of the loss minimization, and then back-propagate through this closed-form solution to get the gradient of the neural network parameters. This is a novel theoretical and practical contribution.

The sections that describe the theory and algorithm behind the approach (sections 3.3 and Appendix A, B, C) are very clear and well written.

The method is compared on a variety of tabular datasets : small (~500 instances ~10 features) and large (20K instances 80 features). This demonstrates that the proposed algorithm scales.

**Weaknesses:**

## The MEM

The main motivation behind the proposed architecture is that the Marginal Effect at the Mean (MEM)\
$$ \frac{\partial f(x)}{\partial x\_i}\bigg\vert\_{x=\bar{x}}$$
is a useful measure for interpreting the model. Indeed, the main motivation behind architecture is that the learned coefficients $\beta$ will coincide with the MEM. Hence, the merits of the MEM should be discussed in more depth. For instance, why should we use this measure of feature importance instead of the mean derivative  $\mathbb{E}\big[\frac{\partial f(X)}{\partial x\_i}\vert\_{x=X}\big]$ or the average derivative amplitude
$\mathbb{E}\big[\big|\frac{\partial f(X)}{\partial x\_i}\vert\_{x=X}\big|^2\big]$ as proposed in [1].

## Choice of Synthetic Data

The ground-truth functions used in the synthetic experiments all take the form $y(x)=\beta_0 + \sum_{i=1}^d\beta_i x_i(1 + f_{-i}(x_{-i}))$, which is the exact form of the NIMO model. This explains why NIMO performs better than competing methods Table 1. It would be fairer to compare NIMO with others architectures on toy models that take a variety of forms e.g. $y(x) = x_1^2 + x_2^2 + x_3^2$ . I would expect NIMO to struggle in this example because the derivative is null at the origin. NIMO cannot perfectly model this function because enforcing a null derivative at the origin requires $\beta_i=0$ for all indices $i$. So, NIMO must output a constant prediction $f(x)=\beta_0$. If NIMO can find an adequate estimate of $y(x) = x_1^2 + x_2^2 + x_3^2$, it would be interesting to plot its behavior near the origin.

## Other Hybrid Architectures

Alternative hydrid architectures have been proposed to enable partial interpretability. Mixture of experts [2] employ a neural network to infer weights $w_k(x)$ that balance the decisions of various experts $e_k(x)$  (e.g. linear models). The model takes the form $f(x) = \sum_{k=1}^K w_k(x) e_k(x)$ and offers local interpretability : when $w_k(x)$ is close to 1 the expert $e_k(x)$ is responsible for the classification of x.
NODE-GAM [3] restrict the network to take the form $f(x) = \sum_{i=1}^d f_i(x_i) + \sum_{i\lt j} f_{ij}(x_{ij})$, which enables interpretability by plotting the main effects $f_i$ and pair-wise interactions $f_{ij}$.

Discussing these articles in the related work would strengthen the need for the novel NIMO architecture. Notably, how is reporting the MEM (which is the build-in feature importance of NIMO) better or worse than reporting the coefficients of the local experts in a Mixture of Experts, or reporting the main effects and pair-wise interactions in a NODE-GAM?


[1] I.M. Sobol’, S. Kucherenko. (2009) "Derivative based global sensitivity measures and their link with global sensitivity indices".
Mathematics and Computers in Simulation, 79(10), 3009-3017.

[2] Ismail, Aya Abdelsalam, et al. "Interpretable Mixture of Experts." Transactions on Machine Learning Research.

[3] Chang, C. H., Caruana, R., & Goldenberg, A. NODE-GAM: Neural Generalized Additive Model for Interpretable Deep Learning. In International Conference on Learning Representations.

**Questions:**

In Figure 5, why are plain Neural Networks worse than LassoNet and NIMO on superconductivity? This result is surprising since NNs are in theory more expressive than either of these two other models. Is it because there is some overfitting going on? Were the hyperparameters for NN chosen to maximize validation set performance? Appendix D.1 describes the hyperparameters used, but not why these values were chosen.

At line 413, it is states that "[Figure 6 reflects] NIMO’s ability to capture nonlinear feature interactions that Lasso cannot represent." I am not sure how to interpret this statement along-side the Figure 6. This is because some features are considered important according to one method and not the other. In diabetes, Lasso considers s1 important while NIMO does not. For Boston, NIMO considers age important, while Lasso does not. Hence it is not immediately apparent that NIMO always captures something that Lasso does not.
Also, can the MEM really be used to make statements about "feature interactions"? A feature could have a large $\beta_i$ but still not interact with other features if its perturbation network has a negligible output $f_{-i}(x_{-i})=0$? To provide evidence for feature interactions, it could be useful to report the strength of the coefficient perturbation $|(1+f_{-i}(x_{-i}))|$ for all features.

Enforcing coefficient sparsity is relevant in lasso because it allows for feature selection. However, in NIMO, a coefficient $\beta_i=0$ does not imply that the model is not relying on $x_i$ to predict. Indeed, $x_i$ is still used in the networks that modify the weights of other features. Given this observation, why is inducing sparsity of $\beta$ in NIMO (via adaptive ridge regression) desirable in the first place? Are there ways sparsity in NIMO could potentially be used for feature selection?

---

> ### Author Response · Authors · 2025-11-19
>
> We would like to thank the reviewer for the valuable feedback, for suggesting further related work and for suggesting further evaluation, which we performed and discuss below. As part of the rebuttal we tried to address all the raised concerns either in text or with new evaluations.
>
> ## Weaknesses:
> > ### The MEM
> For instance, why should we use this measure of feature importance instead of the mean derivative $\mathbb E [ \frac{\partial f(X)}{\partial x_i} \vert_{x=X} ]$ or the average derivative amplitude $\mathbb E [| \frac{\partial f(X)}{\partial x_i}|_{x=X} |^2]$ as proposed in [1].
>
> We thank the reviewer for raising this important question regarding why we focus on the Marginal Effect at the Mean (MEM) rather than alternative notions of global feature importance such as the mean derivative or the average derivative amplitude. The distinction between MEM and other aggregated marginal effects is widely discussed in the interpretability literature (e.g., [2]). Our choice of MEM is motivated by several considerations:
> 1. Different global derivative measures emphasize different aspects of the function, and there is currently no consensus on a universally preferred global importance metric for nonlinear models. Statistics such as the mean derivative, or its squared amplitude, average heterogeneous local effects over the input distribution, which may obscure important nonlinearities or interaction patterns.
> 2. The mean derivative and derivative-amplitude measures are inherently post-hoc summaries, and as such do not correspond to identifiable global parameters of the model. They can be computed for any model but do not themselves induce a stable, population-level interpretability structure.
> 3. A central objective of our architecture (NIMO) is to introduce explicit, model-internal interpretability through the coefficients $\beta$. The design ensures that $\beta_i$ coincides exactly with the MEM, giving it a direct and transparent meaning analogous to regression coefficients in linear models. This structural alignment would not hold if we were to target other derivative-based summaries, which depend on the data distribution rather than on a model-encoded parameter.
>
> [1] I.M. Sobol’, S. Kucherenko. (2009) "Derivative based global sensitivity measures and their link with global sensitivity indices". Mathematics and Computers in Simulation, 79(10), 3009-3017.
>
> [2] Scholbeck, Christian A., et al. "Marginal effects for non-linear prediction functions." Data Mining and Knowledge Discovery 38.5 (2024): 2997-3042.
>
> > ### Choice of Synthetic Data
> It would be fairer to compare NIMO with others architectures on toy models that take a variety of forms e.g. $y(x) = x_1^2 + x_2^2 + x_3^2$. I would expect NIMO to struggle in this example because the derivative is null at the origin. NIMO cannot perfectly model this function because enforcing a null derivative at the origin requires $\beta_i = 0$ for all indices $i$. So, NIMO must output a constant prediction $f(x) = \beta_0$. If NIMO can find an adequate estimate of $y(x) = x_1^2 + x_2^2 + x_3^2$, it would be interesting to plot its behavior near the origin.
>
> We thank the reviewer for this insightful observation.
> 1. We designed the synthetic experiments primarily to showcase that NIMO works as expected and it can actually learn the correct linear structure and nonlinear corrections. For this reason we built examples with precisely this structure. By a design choice (further motivated in the global response), NIMO’s nonlinear correction term $g_{u_j}(\cdot)$ does not receive $x_j$ as input (a necessary architectural constraint to ensure global interpretability), since it guarantees that $\text{MEM}_j = \beta_j$. Consequently, NIMO cannot represent purely self-nonlinear terms such as $x_j^2$, and will indeed collapse to a constant model in such cases.
> 2. In the revised version (Sec. 4.4 and Appendix D.6), we experimentally verify this limitation on the suggested quadratic example. However, we also demonstrate that this issue can be addressed through simple feature engineering, such as polynomial basis expansion, as long as the transformed features remain interpretable. When the squared terms are included explicitly as features, NIMO successfully recovers the correct relationships while retaining meaningful global interpretability. The interpretability of highly nonlinear datasets is ill-posed according to our definition and goes beyond the scope of our paper.

---

> > ### Author Response · Authors · 2025-11-19
> >
> > > ### Other Hybrid Architectures
> > Alternative hydrid architectures have been proposed to enable partial interpretability. Mixture of experts [2] employ a neural network to infer weights $w_k(x)$ that balance the decisions of various experts $e_k(x)$ (e.g. linear models). The model takes the form $f(x) = \sum_{k=1}^K w_k(x) e_k(x)$ and offers local interpretability : when $w_k(x)$ is close to 1 the expert $e_k(x)$ is responsible for the classification of x. NODE-GAM [3] restrict the network to take the form $f(x) = \sum_{i=1}^d f_i(x_i) + \sum_{i<j}f_{ij}(x_{ij})$, which enables interpretability by plotting the main effects $f_i$ and pair-wise interactions $f_{ij}$.
> > Discussing these articles in the related work would strengthen the need for the novel NIMO architecture.
> >
> > We thank the reviewer for pointing out these related hybrid architectures. Mixture-of-Experts and NODE-GAM are indeed important approaches that combine neural networks with interpretable components. In the revised version, we have added a brief discussion of these works in the Related Work section (Sec. 2), highlighting how they differ from NIMO. While these models provide various forms of local or component-wise interpretability, they do not offer a global, sample-independent coefficient interpretation analogous to NIMO’s $\beta$, which remains a key distinguishing feature of our architecture. Furthermore, NIMO still allows for local interpretability.
> >
> > > Notably, how is reporting the MEM (which is the build-in feature importance of NIMO) better or worse than reporting the coefficients of the local experts in a Mixture of Experts, or reporting the main effects and pair-wise interactions in a NODE-GAM?
> >
> > As discussed in our global response, interpretability can be framed from multiple perspectives, and different interpretable machine learning models serve different purposes. We do not view MEM-based interpretation as universally “better’’ or “worse’’ than the explanations provided by Mixture-of-Experts or NODE-GAM; rather, the choice depends on the objectives and the questions one aims to answer.
> > 1. Different interpretability goals require different summaries.
> > In many applications, a purely local, per-instance explanation is sufficient. However, in others, such as medical or policy decision-making, it is equally important to have a **global**, sample-independent summary of how each feature influences outcomes across the population. For example, a local explanation can answer: Given this specific patient’s age and clinical features, how would a small change in age affect their individual risk? A global explanation, on the other hand, answers: Holding all other features constant, how does age affect disease risk at the population level? NIMO is designed to provide **both**: the intelligible global interpretation via marginal effects at the mean (MEM), and per-instance explanation through the network.
> > 2. These approaches are not mutually exclusive.
> > In fact, NIMO can be incorporated into a Mixture-of-Experts framework, where each expert is itself a NIMO model. This would provide even finer-grained explanations across sub-populations while retaining the global interpretability of the $\beta$ coefficients for each sub-population within each expert. In fact, this is also an ongoing research for us.
> >
> > ## Questions:
> > > In Figure 5, why are plain Neural Networks worse than LassoNet and NIMO on superconductivity? This result is surprising since NNs are in theory more expressive than either of these two other models. Is it because there is some overfitting going on? Were the hyperparameters for NN chosen to maximize validation set performance? Appendix D.1 describes the hyperparameters used, but not why these values were chosen.
> >
> > We thank the reviewer for pointing this out. After carefully re-examining our experimental setup, we found that the suboptimal performance of the plain neural network on the superconductivity dataset was caused by an overly strong dropout rate. Most hyperparameters were selected by maximizing validation performance, and for the relatively small Diabetes and Boston Housing datasets, strong dropout (p=0.6) was indeed necessary to avoid overfitting. However, we inadvertently applied the same level of dropout to the superconductivity dataset, which is substantially larger. In this setting, the strong dropout caused underfitting, leading to the suboptimal performance. After revisiting the hyperparameter selection for the superconductivity dataset, we found that a smaller dropout rate (p=0.1) is more appropriate given the dataset size. With this adjustment, the neural network’s performance is at par with NIMO, as expected. Figure 5 has been updated in the revised version to reflect the corrected results.

---

> > > ### Author Response · Authors · 2025-11-19
> > >
> > > > At line 413, it is states that "[Figure 6 reflects] NIMO’s ability to capture nonlinear feature interactions that Lasso cannot represent." I am not sure how to interpret this statement along-side the Figure 6. This is because some features are considered important according to one method and not the other. In diabetes, Lasso considers s1 important while NIMO does not. For Boston, NIMO considers age important, while Lasso does not. Hence it is not immediately apparent that NIMO always captures something that Lasso does not.
> > >
> > > We thank the reviewer for pointing out this ambiguity. We would like to clarify what we intended:
> > > 1. The observed differences in feature importance between Lasso and NIMO can arise from factors beyond nonlinear interactions. In particular, both datasets exhibit strong correlations among certain features. In such cases, feature selection in Lasso, and similarly in NIMO, can become unstable, as multiple correlated features may explain the response equally well.
> > > 2. For the Diabetes dataset, the variance for the coefficients of s1 and s2 is quite large, indicating randomness in the feature selection. In such cases, Lasso can select a different but equally predictive feature from the same correlated group.
> > > 3. Therefore, discrepancies in which features are marked as important by Lasso and NIMO should not be interpreted solely as evidence of nonlinear interactions; they may simply reflect different ways in which the two models handle correlated features. In the revised version of the pdf, we have modified our phrasing.
> > >
> > > > Also, can the MEM really be used to make statements about "feature interactions"? A feature could have a large $\beta_i$ but still not interact with other features if its perturbation network has a negligible output $f_{-i}(x_{-i})=0$? To provide evidence for feature interactions, it could be useful to report the strength of the coefficient perturbation $|1+ f_{-i}(x_{-i})|$ for all features.
> > >
> > > We agree that MEM alone cannot be used to infer the presence of feature interactions. To clarify this point, and following the reviewer’s suggestion, we now report and compare the strength of the coefficient perturbations, measured as $|1+ g_{u_j}(x_{-j})|$ for each feature in the revised version (Appendix D.8, Figure 20). The results show that, for the Diabetes dataset, these values remain very close to 1 across all features, indicating that the neural correction terms contribute minimally and that the underlying structure is largely linear. In contrast, for the Boston Housing dataset, the perturbation magnitudes deviate substantially from 1, providing clear evidence of stronger nonlinear interactions.
> > >
> > >
> > > > Enforcing coefficient sparsity is relevant in lasso because it allows for feature selection. However, in NIMO, a coefficient $\beta_i = 0$ does not imply that the model is not relying on $x_i$ to predict. Indeed, $x_i$ is still used in the networks that modify the weights of other features. Given this observation, why is inducing sparsity of $\beta$ in NIMO (via adaptive ridge regression) desirable in the first place? Are there ways sparsity in NIMO could potentially be used for feature selection?
> > >
> > > We thank the reviewer for raising this important question.
> > > 1. The fact that a feature $x_i$ may still appear in the networks even when $\beta_i = 0$ is intentional. This design allows NIMO to model cross-feature nonlinear interactions: removing $x_i$ entirely from the model would eliminate these interactions and significantly reduce the expressive capacity of the model (basically reducing it to a linear model).
> > > 2. Inducing sparsity in $\beta$ remains desirable because $\beta$ encodes the **global, sample-independent marginal effect at the mean (MEM)**. When examining MEM, all local nonlinear corrections vanish, and the interpretability reduces to the global coefficients $\beta_i$. Sparse $\beta$ therefore yields a concise global explanation of which features exert population-level effects.
> > > 3. Sparsity in $\beta$ alone is not sufficient for full feature selection in NIMO. To determine whether a feature meaningfully contributes to the prediction, one should also examine the magnitude of its coefficient perturbations $\lvert 1 + g_{u_j}(x_{-j}) \rvert$. A feature with $\beta_i = 0$ and perturbation values consistently close to 1 across the dataset can be safely regarded as non-contributory and removed.

---

> ### Comment · Reviewer_F6aQ · 2025-11-25
> **Reviewer Response**
>
> Thank you very much for answering my questions, introducing a new toy example, correcting the MLPs on Conductivity, and updating the manuscript. I have accordingly increased my score.
>
> I am still not 100% convinced about the merits of the MEM as a global Marginal effects for non-linear prediction functions.
> In the toy example $f(x) = x_1^2 + x_2^2 + x_3^2$, the MEM is null which suggests the global explanation: no feature is important, That is simply false. Yet, by reading the reference [1] shared by the authors in the rebuttal, I have to concede that the MEM is a relevant quantity studied in the Statistics literature (a literature parallel to the XAI literature). Thus, to better motivate the MEM, I would recommend to add citation [1] to Section 3.2 and add a small paragraph that acknowledges  the flaws of this metric. I think that citation [1] is more convincing that [2] (the one given in the paper) since [2] is a general book on statistics while [1] is published work that compares different Marginal Effects (the MEM and others).
>
> [1] Scholbeck, Christian A., et al. "Marginal effects for non-linear prediction functions." Data Mining and Knowledge Discovery 38.5 (2024): 2997-3042.
>
> [2] Mike Nguyen. A Guide on Data Analysis. Bookdown, 2020. URL https://bookdown.org/mike/data_analysis/.

---

> > ### Author Response · Authors · 2025-11-27
> >
> > We thank the reviewer for the constructive follow-up and for increasing the score. We agree that MEM has known limitations when applied to arbitrary nonlinear functions, especially in the presence of strong nonlinearities, as discussed in [1]. However, NIMO is not designed to model arbitrary nonlinearities of that form. Instead, its architecture explicitly separates a global linear component (captured by $\beta$) from local nonlinear adjustments. This structural constraint ensures that the global linear effect remains identifiable and meaningful, and therefore the MEM remains an appropriate and interpretable global effect measure within this model class, even though it may be unsuitable for functions with significant self-interacting terms. In the specific case mentioned, our solution of using a polynomial basis expansion would provide a useful interpetaion under MEM (given that the transformed features are still interpretable).
> >
> > Following the reviewer’s suggestion, we have updated Section 3.2 in the revised manuscript to include (1) a citation to Scholbeck et al. [1] and (2) a brief paragraph acknowledging these general limitations of MEM and clarifying why, given NIMO’s architectural design and modeling scope, MEM is still applicable and interpretable in our framework.

---

### Official Review · Reviewer_cUUF · 2025-10-31

**Soundness:** 3
**Presentation:** 3
**Contribution:** 3
**Rating:** 6
**Confidence:** 3

**Summary:**

Neural Networks are powerful but not inherently interpretable. Linear probes are interpretable but lack expressiveness. The authors aim to bridge the gap by introducing a new model - NIMO. NIMO is motivated by the marginal effect $ME_j$, which measures how much a prediction $f(x)$ changes when changing the value of $x_j$. Specifically the metric marginal effect at the mean $MEM_j$ factorizes out nicely, so that under the mean the marginal effect $MEM_j = \beta$, the fixed and learned linear coefficient. For the optimization of the model they utilize a profile likelihood approach which reduces the problem, so that only the parameters $u$ need to be trained. Furthermore they extend the idea to generalized linear models. In S3.4 the authors demonstrate that Nimo is able to retrieve accurate linear coeffcients $\beta$ and that $g_{u_j}(x_{-j})$ is accurately predicting the nonlinear scalars of feature $f_j$. In section 4.2 H1 the authors demonstrate that in 3 synthetic low data settings their method outperforms competitors in MSE. In 4.2 H2 they showcase how Nimo accurately captures the linear coeffcients in a vanilla regression setting and in 3 non linear settings which are further explained in the appendix. Sec 4.2 H3 compares a) the performance of Nimo on real world datsets (diabetis, housing and superconductivity) in which they perform competitive and b) how the shap feature contributions align with the NIMO coefficients.

in 4.3 the authors run ablation studies regarding the design of $g_u$

In general Nimo contributes a novel model architecture and training recipe to enable interpretable by design and yet powerful AI models.

**Strengths:**

- The authors take the definition of MEM seriously and introduce a novel and clever method that enhances MEM interpretability.
- The Introduction, as well as Sections 3.1 and 3.2, are particularly clear and easy to follow.
- NIMO can model non-linear relationships while providing linear interpretations at any sample x.
- The toy model in Section 3.4 is excellent for illustrative purposes.
- NIMO is competitive with other interpretable models in terms of performance, while offering intriguing properties for feature ranking.

**Weaknesses:**

* The limitations of this model are not analyzed in great detail. I appreciate the analysis of what this Nimo can do equally as much as the discussion of the limitations. However the limitations are only discussed in a single sentence. Addressing these problems convincingly e.g. by experiments that clearly display key limitations, is crucial imo.
* The authors themselves acknowledge the importance of both local and global explanations, yet they do not provide any local explanations, but only global explanations.
* No code is available, severely diminishing means of reproduction.
* Minor things: In line 414 it would be great to link to the Figure in the Appendix

**Questions:**

- Did the authors evaluate the statistics of $\beta_i + g_{u_j}(x_{-j})$? It would be interesting to find out whether the degree of non-linearity can be assessed through such statistics. For example, if the impact of $f_i$ were bi-modal, $\beta_i$ might be close to zero. I would be particularly interested in the differences between the Diabetes and Boston Housing datasets, as the authors state that the Diabetes dataset exhibits more linear interactions. It would be great if this were directly reflected in the statistics of the effective contributions.

- I would like to see analysis of scenarios in which features do not contribute linearly e.g. on the xor problem. On of my concerns is that Nimo would still be able to solve the problem, but would not be faithful in its $\beta_i + g_{u_j}(x_{-j})$ score. Is it possible to give guarantees to empirical bounds to the faithfulness of the feature contribution?

---

> ### Author Response · Authors · 2025-11-19
>
> We would like to thank the reviewer for the encouraging feedback and for proposing further evaluation of the limitation and the role of linear and nonlinear contributions. As part of the rebuttal we tried to address all the raised concerns either in text or with new evaluations.
>
> ## Weaknesses:
> > The limitations of this model are not analyzed in great detail. I appreciate the analysis of what this Nimo can do equally as much as the discussion of the limitations. However the limitations are only discussed in a single sentence. Addressing these problems convincingly e.g. by experiments that clearly display key limitations, is crucial imo.
>
> We agree with the reviewer that the limitations of NIMO deserve a more thorough analysis. We included a more detailed discussion in the global rebuttal and we wrote a dedicated “Limitations” subsection in Sec. 4.4. The limitation originally discussed in the paper is that NIMO can struggle with highly nonlinear datasets or features with self-interactions (e.g., $x_j f(x_j)$). We now inlcuded an additional synthetic experiment where the data generating process does **not** match the NIMO form and propose a solution (see Appendix D.6).
>
> > The authors themselves acknowledge the importance of both local and global explanations, yet they do not provide any local explanations, but only global explanations.
>
> NIMO provides both global and local explanations by construction.
> 1. The global explanation is given by the coefficient vector $\beta$, which corresponds to the Marginal Effect at the Mean (MEM) and offers a stable population-level summary.
> 2. The local explanation for each feature $j$ at an instance $x$ is captured by the locally adjusted coefficient $s_j (x) = \beta_j (1 + g_{u_j}(x_{-j}))$, with the corresponding local contribution $c_j(x) = x_j s_j(x)$. This quantity reflects how the impact of feature $j$ varies across instances. In Figure 3 our toy example already showcases that the nonlinear correction terms are learned accurately, which confirms NIMO’s ability to provide meaningful local explanations.
> 3. In the revised version, we now include a dedicated subsection in Appendix D.7 that explicitly presents local explanations on both synthetic and real datasets, illustrating NIMO’s instance-specific interpretability in a clear and systematic manner. For a more detailed discussion about global and local interpretability we refer to the global response.
>
>
> > No code is available, severely diminishing means of reproduction.
>
> The full implementation of NIMO, including training and evaluation scripts to reproduce all experiments, is provided in the Supplementary Material (as part of the attached code archive). We will also release the code as a public repository upon acceptance to further facilitate reproducibility.
>
> > Minor things: In line 414 it would be great to link to the Figure in the Appendix
>
> We thank the reviewer for this suggestion. In the revised version of the paper we have added the link to the corresponding figure in the Appendix.

---

> > ### Author Response · Authors · 2025-11-19
> >
> > ## Questions:
> > > Did the authors evaluate the statistics of $\beta_i + g_{u_j}(x_{-j})$? It would be interesting to find out whether the degree of non-linearity can be assessed through such statistics. For example, if the impact of $f_i$ were bi-modal, $\beta_i$ might be close to zero. I would be particularly interested in the differences between the Diabetes and Boston Housing datasets, as the authors state that the Diabetes dataset exhibits more linear interactions. It would be great if this were directly reflected in the statistics of the effective contributions.
> >
> > Thank you for the insightful suggestion. This question is closely related to Reviewer F6aQ’s comment regarding how to quantify the degree of nonlinear interactions. Following the reviewers’ feedback, we agree that examining statistics derived from the correction terms can help reveal how nonlinear each feature’s behavior is in different datasets. Rather than using the additive form $\beta_i + g_{u_j}(x_{-j})$ (which does not correspond directly to NIMO’s architecture), we evaluate the quantity $\lvert 1 + g_{u_j}(x_{-j}) \rvert$, which directly captures the magnitude of the instance-specific nonlinear correction applied to feature $j$.
> > In the revised version (Appendix D.8), we include a new subsection presenting these statistics for both the Diabetes and Boston Housing datasets. The results clearly show that the Diabetes dataset exhibits correction values much closer to 1 across all features, indicating weak nonlinearity, while the Boston Housing dataset shows substantially larger deviations, consistent with our claims in the main paper.
> >
> > > I would like to see analysis of scenarios in which features do not contribute linearly e.g. on the xor problem. On of my concerns is that Nimo would still be able to solve the problem, but would not be faithful in its $\beta_i + g_{u_j}(x_{-j})$ score. Is it possible to give guarantees to empirical bounds to the faithfulness of the feature contribution?
> >
> > We thank the reviewer for raising this important point. It is indeed necessary to analyze scenarios where features do not contribute linearly. As discussed earlier and in our general response, NIMO is not designed to address highly nonlinear datasets. To illustrate this limitation concretely, we include in Sec. 4.4 and Appendix D.6 the example suggested by reviewer F6aQ, $y(x) = x_1^2 + x_2^2 + x_3^2$. This allows us to experimentally demonstrate the limitation and, at the same time, introduce a simple remedy through basis expansion.
> > Our results show that while NIMO may fail to provide a globally linear + locally nonlinear explanation in its original form, applying an appropriate basis expansion allows the method to recover the correct structure. This is not a universal solution; it relies on the transformed features remaining interpretable (e.g., $x_1^2$ is still meaningful). Datasets requiring highly complex or non-interpretable transformations fall outside the scope of our interpretability framework, and NIMO is not intended for such settings.

---

> > > ### Author Response · Authors · 2025-11-27
> > >
> > > We would like to thank again the reviewer for the encouraging comment and constructive suggestions. As the end of the rebuttal is approaching, we would appreciate knowing whether the concerns have been adequately addressed, or if there are still some issues remaining that we can further address to improve our work.

---

### Author Response · Authors · 2025-11-19
**Global response**

We thank all reviewers for their time and valuable feedback. We appreciate some potential misunderstanding that were raised and the suggestion for further evaluation, which we provided as part of the rebuttal. We would also like to thank reviewers for acknowledging the strengths of NIMO as a powerful yet interpretable-by-design model.

In order to address common concerns and to provide a summary to the AC, we provide global response below. We further provide answer to each individual reviewer as individual comments. Below, we clarify the key points regarding our model, including interpretability and limitations.

**Interpretability and the role of MEM**

Depending on the specific questions of interest, interpretability can be framed from different perspectives and at different levels of granularity. Within our work, our goal is to preserve a **global, population-level** interpretation in the form of the Marginal Effect at the Mean (MEM), while still allowing **local, instance-specific** nonlinear explanations.
1. *Global-interpretability*. A key property of our architecture is that the linear coefficients $\beta$ are input-independent parameters, and coincide *by design* with the MEM: $MEM_j = \frac{\partial f(x)}{\partial x_j}|_{x=\bar x} = \beta_j$. This gives NIMO the same type of global interpretability as linear regression models.
2. *Local-interpretability*. At the same time, the correction term $1+g_{u_j}(x_{-j})$ provide nonlinear corrections that depend on the input, hence local. This is done also by design and requires the nonlinear term $g_{u_j}(x_{-j})$ not to depend on $x_j$, namely $x_{-j}$. If this was not the case the nonlinear term could explain the linear part as well, while we avoid this by construction, thus preserving the global interpretability.

**NIMO applicability**

In NIMO the nonlinear correction term $g_{u_j}(x_{-j})$ for the feature $j$ can take any feature as input besides $j$ itself (i.e. $x_{-j}$). This design choice is needed to maintain the interpretability of linear coefficients as the MEM, which is the main objective of NIMO. Therefore, in specific settings where the underlying nonlinearity contains self-interacting terms, e.g. $x_j \cdot f(x_j)$, NIMO would fail to learn the exact underlying dynamics; i.e. the linear coefficients will be incorrect. However, in many cases non-linear features $f(x_j)$ are related to standard basis expansions. Using basis expansion, NIMO will learn the correct model while providing intepretability in terms of the feature $f(x_j)$. However, in some cases $f(x_j)$ is not associated with an interpretable basis or the basis/interaction is not known at all. In such cases, NIMO fails; but we would argue that interpretability in such settings, as we define it, is inherently difficult independently of our model. Therefore, we consider such cases to be beyond the scope of our paper. In the revised version of the paper we included a dedicated limitation section (Sec 4.4) and we included an additional experiment (Appendix D.6) where we showcase precisely this limitation and how it can be overcome in practice if knowledge about the (interpretable) feature expansion is available.

We highlighted in red the changes made in the revised pdf.

---

### Author Response · Authors · 2025-12-03
**Short summary of the rebuttal**

We thank the new AC for their effort in reassessing our submission under the current circumstances. Below we provide a brief, factual summary of the status of our rebuttal and reviewer interactions prior to the incident.

All rebuttal exchanges, discussions, and the subsequent score updates occurred before the incident was known publicly.

1. Reviewer **cUUF** (6) provided a positive assessment of the paper highlighting the novelty and the advantage of having linear interpratable coefficients together with nonlinear interpretable corrections. The reviewer asked for further clarifications on the limitations and raised minor concerns, which we addressed in the rebuttal.
2. Reviewer **F6aQ** (4->6) raised some concerns regarding the motivation for MEM and asked for further synthetic experiments. In the rebuttal we provided a detailed motivation and a new explanatory toy experiment. Reviewer F6aQ found our clarifications and the toy example helpful and increased the score to 6.
3. Reviewer **DuVs** (4) raised some questions concerning global and local interpretability and asked for further benchmarking. In the rebuttal we described in details how NIMO is able to provide both interpretabilities by design. We also benchmarked NIMO on 10 additional datasets from the suggested PMLB. We addressed also other minor concerns in detail. However, reviewer DuVs did not engage further in the discussion and did not specify which concerns remain open.

We hope this summary helps contextualize the current state of the reviews. We appreciate your time and attention.

---

### Meta-Review · Area_Chair_68DG · 2026-01-07

**Summary:**

Reviewers agree that the paper addresses an important problem in interpretable machine learning and presents a technically sound and novel model that combines global linear coefficients with nonlinear corrections learned by a neural network. The optimization strategy, which analytically eliminates the linear parameters and backpropagates through the closed-form solution, was viewed as a clear technical contribution. Concerns focused primarily on the interpretability claims—particularly the justification of marginal effects at the mean (MEM) as a notion of global interpretability—the scope of model limitations, and the breadth of empirical validation. Overall, the reviews reflect a borderline but positive assessment following rebuttal.

**Reviewer Concerns:**

Several reviewer concerns were substantively addressed in the rebuttal. The authors clarified the distinction between global and local interpretability, expanded the discussion of limitations and failure modes, corrected experimental issues, added further empirical evaluations, and improved the positioning relative to related hybrid interpretable models. These clarifications resolved a number of misunderstandings raised in the initial reviews.

However, some concerns remain outstanding. One reviewer remains unconvinced that the proposed notion of global interpretability is sufficiently compelling in general nonlinear settings and questions whether sparsity in the global coefficients necessarily corresponds to meaningful feature relevance when nonlinear correction networks remain active. This reviewer maintained their score after rebuttal, indicating continued reservations about the interpretability framing rather than the technical correctness of the method.

**Reviewer Scores:**

Reviewer cUUF: Likely remains at 6 (marginally above acceptance threshold).

Reviewer F6aQ: Increased to 6 after rebuttal and would likely remain there.

Reviewer DuVs: Likely remains at 4, as the reviewer explicitly maintained their score.

---

### Decision · Program_Chairs · 2026-01-26

Accept (Poster)